# Synoptic-scale controls of fog and low cloud variability in the Namib Desert

Hendrik Andersen[1,2], Jan Cermak[1,2], Julia Fuchs[1,2], Peter Knippertz[1], Marco Gaetani[3,4], Julian Quinting[1], Sebastian Sippel[5,6], and Roland Vogt[7]

[1]Karlsruhe Institute of Technology (KIT), Institute of Meteorology and Climate Research, Karlsruhe, Germany
[2]Karlsruhe Institute of Technology (KIT), Institute of Photogrammetry and Remote Sensing, Karlsruhe, Germany
[3]LISA/IPSL, CNRS, Université Paris Est Créteil, Université Paris, Créteil, France
[4]LATMOS/IPSL, CNRS, Sorbonne Université, Université Paris-Saclay, Paris, France
[5]ETH Zurich, Institute for Atmospheric and Climate Science, Zurich, Switzerland
[6]Norwegian Institute of Bioeconomy Research, Ås, Norway
[7]University of Basel, Department of Environmental Sciences, Basel, Switzerland

**Correspondence:** Hendrik Andersen (hendrik.andersen@kit.edu)

**Abstract.** Fog is a defining characteristic of the climate of the Namib Desert and its water and nutrient input are important for local ecosystems. In part due to sparse observation data, the local mechanisms that lead to fog occurrence in the Namib are not yet fully understood, and to date, potential synoptic-scale controls have not been investigated. In this study, a recently established 14-year data set of satellite observations of fog and low clouds in the central Namib is analyzed in conjunction with reanalysis data to identify synoptic-scale patterns associated with fog and low-cloud variability in the central Namib during two seasons with different spatial fog occurrence patterns. It is found that during both seasons, mean sea level pressure and geopotential height at 500 hPa differ markedly between fog/low-cloud and clear days, with patterns indicating the presence of synoptic-scale disturbances on fog and low-cloud days. These regularly occurring disturbances increase the probability of fog and low-cloud occurrence in the central Namib in two main ways: 1) an anomalously dry free troposphere in the coastal region of the Namib leads to stronger longwave cooling of the marine boundary layer increasing low-cloud cover, especially over the ocean where the anomaly is strongest, and 2) local wind systems are modulated, leading to an onshore anomaly of marine boundary-layer air masses. This is consistent with air mass backtrajectories and a principal component analysis of spatial wind patterns that point to advected marine boundary-layer air masses on fog and low-cloud days, whereas subsiding continental air masses dominate on clear days. Large-scale free-tropospheric moisture transport into southern Africa seems to be a key factor modulating the onshore advection of marine boundary-layer air masses during April, May, and June, as the associated increase in greenhouse gas warming and thus surface heating is observed to contribute to a continental heat low anomaly. A statistical model is trained to discriminate between fog/low-cloud and clear days based on information on large-scale dynamics. The model accurately predicts fog and low-cloud days, illustrating the importance of large-scale pressure modulation and advective processes. It can be concluded that Namib-region fog is predominantly of advective nature, and that fog and low-cloud cover is effectively maintained by increased cloud-top radiative cooling. Seasonally different manifestations of synoptic-scale disturbances act to modify its day-to-day variability and the balance of mechanisms leading to its formation

and maintenance. The results are the basis for a new conceptual model on the synoptic-scale mechanisms that control fog and low cloud variability in the Namib Desert, and will guide future studies of coastal fog regimes.

## 1 Introduction

In moist climates, fog is typically viewed as an atmospheric phenomenon that disturbs traffic systems and negatively affects physical and psychological health (e.g., Bendix et al., 2011). In the hyperarid Namib Desert, however, the water input of fog is key to the survival of many species (e.g., Seely et al., 1977; Seely, 1979; Ebner et al., 2011; Roth-Nebelsick et al., 2012; Warren-Rhodes et al., 2013; Henschel et al., 2018; Gottlieb et al., 2019). Despite this ecological significance, the local mechanisms that control the formation and spatiotemporal patterns of Namib-region fog are not yet fully understood, and potential linkages to synoptic-scale variability have yet to be explored. With regional climate simulations suggesting a warmer and even dryer climate (James and Washington, 2013; Maúre et al., 2018), fog could become an even more essential water source for regional ecosystems in the future. However, the lack of understanding concerning fog and low-cloud (FLC) processes and their interactions with dynamics, thermodynamics, aerosols, and radiation in this region (Zuidema et al., 2016; Formenti et al., 2019) limits the accuracy of and confidence in projected changes of fog patterns (e.g., Haensler et al., 2011).

Field observations of local meteorological parameters and fog have led to the distinction between two main fog types occurring in the region: advection fog and high fog. Advection fog can form when a moist warm air mass is transported over a cool ocean (Gultepe et al., 2007) and has been reported to occur mainly during austral winter, affecting a coastal strip of < 30–40 km (Seely and Henschel, 1998). High fog is described as a low stratus that frequently reaches more than 60 km inland between September and March and leads to fog where the advected stratus base intercepts the terrain (Seely and Henschel, 1998). While the two fog types are reported to be transported inland with different wind systems (for a review of local wind systems see Lindesay and Tyson (1990)), they are both described to be of advective nature. In Olivier and Stockton (1989), a coastal low is described as the mechanism that, in case of a narrow coastal upwelling region, drives the onshore advection of foggy air masses into the region of Lüderitz in southern Namibia during austral summer, while during winter they find fog to be associated with cold fronts. However, they assume that, while undetected, coastal lows were also present in these cases, as they typically precede the passage of a cold front (Olivier and Stockton, 1989; Reason and Jury, 1990). Recent analyses of diurnal FLC characteristics have shown that the timing of FLC occurrence depends on the distance to the coastline, with FLCs occurring significantly earlier at the coast than further inland, which is an indication for the dominance of advective processes (Andersen and Cermak, 2018; Andersen et al., 2019). Also, measurements of fog microphysics during the AEROCLO-sA field campaign in the Namib suggest that the observed fog events were advected cloudy air masses from the ocean (Formenti et al., 2019). While it has long been acknowledged that other fog types (e.g., radiation fog and frontal fog) can occur in the Namib as well (e.g., Jackson, 1941; Nagel, 1959), many statements regarding fog formation mechanisms in the historical literature

do not seem to be founded on extensive and coherent observational evidence. Until recently, the occurrence of radiation fog, i.e. fog formation near the surface due to local radiative cooling under clear-sky conditions and without advective influence (Gultepe et al., 2007), was seen as a comparably rare situation (e.g., Seely and Henschel, 1998; Eckardt et al., 2013). This was questioned when, based on analyses of stable isotopes of fog water samples, Kaseke et al. (2017) and Kaseke et al. (2018) found that the majority of their collected fog water samples stemmed from sweet water sources and interpreted this as evidence of predominant occurrence of radiation fog. Based on these findings, they postulated a potential shift from advection-dominated fog to radiation-dominated fog in the Namib Desert (Kaseke et al., 2017). Thus, the importance of the various fog formation mechanisms is currently a subject of scientific debate.

The goal of this study is to better understand the synoptic-scale conditions under which Namib-region FLCs occur, how synoptic-scale variability changes local conditions, and thereby to assess the relevance of different potential fog formation mechanisms. To this end, a 14-year time series of geostationary satellite observations of FLCs in the central Namib is combined with reanalysis data and air-mass backtrajectories to systematically analyze the large-scale dynamic conditions and air-mass characteristics that are associated with FLC occurrence in the Namib. The guiding hypothesis for this study is:

Fog and low clouds in the central Namib are primarily of advective nature and therefore associated with distinct synoptic-scale patterns of atmospheric dynamics and air-mass history. Thus, they can be statistically predicted with information on atmospheric circulation.

## 2  Data and methods

### 2.1  Satellite observations of FLCs

The Spinning-Enhanced Visible and Infrared Imager (SEVIRI) sensor, mounted on the geostationary Meteosat Second Generation (MSG) satellites, is ideally suited to provide spatiotemporally coherent observations of clouds. It features a spatial resolution of 3 km at nadir and scans its full disk every 15 min (96 hemispheric scans per day, (Schmetz et al., 2002)). In the context of this study, 14 years (2004–2017) of SEVIRI data are used to continuously detect FLCs with the algorithm developed by Andersen and Cermak (2018). The algorithm relies mostly on a channel difference in the thermal infrared ($12.0$–$8.7\mu m$), and in an extensive validation against surface observations, this technique has shown a good skill (97 % overall correctness of the classification). The 14-year FLC data set used here has already been applied to study spatial and temporal patterns of FLC occurrence along the southwestern African coast in Andersen et al. (2019). It should be noted that this satellite technique does not discriminate between fog and other low clouds.

The focus of this study is on FLCs in the central Namib, from where the majority of historical and present-day station measurements stem (e.g. Nagel, 1959; Nieman et al., 1978; Lancaster et al., 1984; Seely and Henschel, 1998; Kaseke et al., 2017; Spirig et al., 2019). To provide a representative measure of the overall central-Namib FLC cover on a daily basis, FLC occurrence is averaged between 3 UTC and 9 UTC (local time is UTC +2h) in the region between $22°$S and $24°$S and up to 100 km inland. Only pixels with at least a 5 % FLC occurrence frequency in the climatology (as in Andersen et al. (2019)) are used. A specified averaging time period is needed to avoid statistically mixing two separate FLC events occurring on successive

nights which would be the case in a daily average FLC occurrence data set. The specific time period is chosen to include all periods of the diurnal cycle, with FLC occurrence rising, peaking, and starting to dissipate (Andersen and Cermak, 2018) during this time. The spatial and temporal averaging is illustrated for an exemplary day in Fig. 1 a). While the specific day shown here is arbitrary, the general feature of maximum FLC cover in the early morning hours and rapid decline shortly after sunrise is typical of the region (Andersen and Cermak, 2018). For further analyses, the data set is divided into 'FLC days' with mean regional FLC cover exceeding 50 % between 3 and 9 UTC, and 'clear days' with mean FLC cover below 3 %. These thresholds are chosen to represent two clearly separated parts of the FLC cover distribution that occur with similar frequencies. The resulting distribution of daily average FLC cover and the number of cases in each class are shown in Fig. 1 b). As the time of sunrise varies by season, the constructed data set is likely to feature a seasonal bias in FLC occurrence. It should be noted that this has no effect on the separation of FLC days and clear days within seasons, the analysis of which is the main purpose of this data set. The resulting monthly average central-Namib FLC cover (Fig. 1 c)) should not be used in a quantitative sense, but rather illustrate the general seasonal cycle of FLCs in this region. It is interesting to note that the seasonal cycle of FLCs is not necessarily coupled to the seasonal cycle of fog occurrence due to the seasonal cycle in the vertical position of the low-cloud layer. For example, at coastal locations of the central Namib fog peaks between April and August (Andersen et al., 2019), while marine fog over the adjacent Atlantic has been found to peak between March and May, with a minimum occurrence between June and August (Dorman et al., 2019).

## 2.2 ERA5 reanalysis

To investigate the large-scale meteorological conditions associated with FLCs in the central Namib, ERA5 reanalysis data from the European Centre for Medium-Range Weather Forecasts (ECMWF) are used. ERA5 is the new generation of reanalysis and follow-up of ERA-Interim (Dee et al., 2011). In comparison to ERA-Interim, it features higher spatial ($0.25°$) and temporal (hourly) resolutions, along with other improvements (Hersbach, 2016).

In the context of this study, 14 years (2004–2017) of meteorological fields are analyzed. To characterize large-scale dynamic and thermodynamic conditions, fields of mean sea level pressure (MSLP), geopotential height at 500, 700, 850, and 925 hPa (Z500, Z700, Z850, Z925), 2 m air temperature (T2m), sea surface temperature (SST), total columnar water vapor (TCWV), specific humidity (Q), as well as winds at 10 m and at all ERA5 pressure levels between 1000 and 500 hPa, and lower tropospheric stability (LTS: computed as the difference between potential temperature at 700 hPa and T2m, (Klein and Hartmann, 1993)) are used. To represent the morning conditions for which FLC is averaged, 6 UTC fields of ERA5 data are selected. While for additional analysis, T2m fields are also used at nighttime (1 UTC and 3 UTC), the 6 UTC fields are used if no specific information on time is given.

## 2.3 Trajectory analysis

24-hour backward trajectories are calculated using the Lagrangian Analysis Tool (LAGRANTO, (Sprenger and Wernli, 2015)). The three wind components needed for the trajectory calculations are taken from ERA5 on a regular $0.5°$ latitude-longitude grid, at 137 model levels in the vertical, and at a 3-hourly temporal resolution. The spatial resolution is used to reduce the

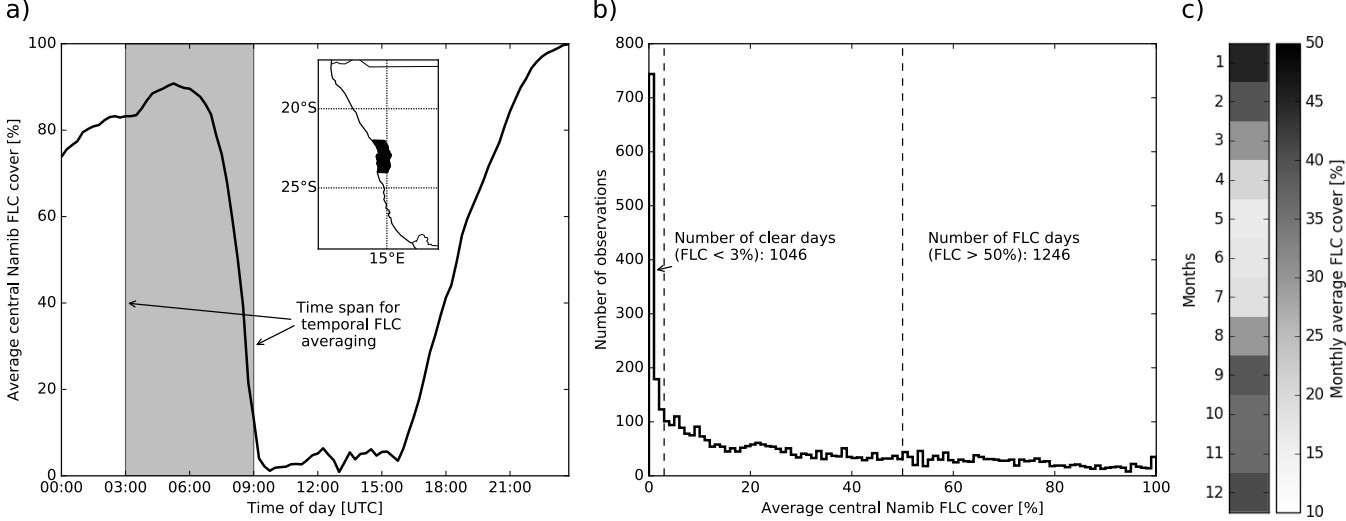

**Figure 1.** a) Illustration of the spatiotemporal averaging for one exemplary day (September 9th, 2015) to create the FLC cover data set. The curve in a) shows the regionally averaged (marked central-Namib region) FLC occurrence, which is then averaged between 3 UTC and 9 UTC (grey area). The resulting average daily morning FLC cover is given in percent. b) Distribution of the resulting central Namib FLC cover for the complete observation period (2004–2017). Observations are separated in two classes: clear days (<3 % mean FLC cover) and FLC days (mean FLC cover >50 %). Days with mean FLC cover between 3 % and 50 % are not considered in analyses based on this classification (2418 cases, for 404 days FLC cover could not be computed due to missing data or complete coverage with higher-level clouds). Panel c) shows monthly averages of the spatiotemporally averaged FLC cover data set.

data volume and computational cost. While the native resolution would be preferable, the general patterns of the trajectories are not expected to be affected, as tests with lower-resolution ERA-Interim data showed comparable results. The trajectories are started daily at 06 UTC for the period April, May, June, September, October and November 2004-2017. Their starting points are located in the central Namib close to Gobabeb at 23°S and 15°E 25 hPa above the surface (at ≈ 940 hPa), which corresponds roughly to 200 m above ground level. By doing so, the back trajectories represent air masses for the levels where fog and low clouds in the region are typically observed (Andersen et al., 2019). In order to obtain insights about the physical properties of the air masses, the temperature, potential temperature, specific humidity, and relative humidity are tracked along the trajectories. The location is chosen, as it is the main site of both historic and present-day scientific activity in the region (Lancaster et al., 1984; Seely and Henschel, 1998; Kaseke et al., 2017; Spirig et al., 2019).

## 2.4 Principal component analysis

The atmospheric variability of the South Atlantic and southern African region is characterized by means of a Principal Component Analysis (PCA, Storch and Zwiers (cf. 1999)). PCA solves the eigenvalues of the data covariance matrix and projects data variability onto an orthogonal basis, i.e. decomposes data variability into independent variability modes. Each mode explains a fraction of the total variance, and is represented by a spatial anomaly pattern and a standardized time series (namely, the

principal components (PCs)) accounting for the amplitude of the anomaly pattern. Here, the PCA is used to analyze daily fields of the zonal and meridional components of 10 m wind at 6 UTC in a domain centered on the Namib (0°–40°E; 40°S–0°N). In the context of this study, the main modes of the wind variability are used to understand possible linkages between atmospheric circulation at the synoptic scale and the daily occurrence of FLCs in the Namib-region. The wind fields are first remapped onto

a 1° regular grid (PCAs are computationally expensive; (Pham-Thanh et al., 2019)), then daily 6 UTC anomalies are computed by subtracting the 14-year climatological average wind components at each grid point. The PCA is applied to the covariance matrix of both components in the domain. Remapping to 1° resolution allows to accurately describe the atmospheric variability at synoptic scale, but smoothing out the variability associated with small-scale effects. The sensitivity of the PCA to the spatial resolution is tested by conducting the analysis based on wind fields remapped to a 2° resolution. The results of the two PCAs

at the different resolutions are very similar, demonstrating their robustness. Daily anomalies are computed with respect to the 14-year sampling of the FLC dataset, in order to compare wind and FLC variability over a homogeneous climatology.

## 2.5    Statistical prediction of FLCs

Statistical modeling of fog or low clouds is typically done by using local fields of a set of predictors, i.e. relevant meteorological fields and aerosol properties (e.g., Andersen et al., 2017; Adebiyi and Zuidema, 2018; Fuchs et al., 2018). The circulation-

induced variability can be captured by spatial patterns of pressure fields (Deloncle et al., 2007; Yu and Kim, 2010; Sippel et al., 2019). A major challenge when using pressure fields (denoted $\mathbf{X}$, as an $n \times p$ matrix of $n$ samples and $p$ predictors located on a grid) to predict a target variable is, however, that the number of (strongly correlated) predictors can quickly outgrow the number of observations. This typically leads to high-variance problems (overfitting) in classical statistical models. The issue can be overcome with shrinkage methods, as e.g. regularized linear models (Hastie et al., 2001). These provide an extension of

linear regression techniques that shrink the regression coefficients of a model by penalizing their size, thereby addressing the aforementioned high-variance issues (Hastie et al., 2001). Ridge regression is a specific example of a regularized linear model where the shrinkage is controlled by a value $\lambda$ that shrinks the coefficients of the model towards zero using the L2 penalty (the squared magnitude of the coefficient value is added as a penalty term to the loss function). This method is well suited for cases with a large number of correlated predictors that are all relevant (coefficients $> 0$) (Friedman et al., 2010). The method can be

used for classification and regression (Friedman, 2012).

Here, the statistical learning method is used in a classification setting. That is, a binary response variable ("FLC day" or "clear day") is modeled using logistic regression regularized with the ridge penalty. In logistic regression with a binary response variable, the "odds ratio" ($log \frac{Pr(FLC\ day|X=x)}{Pr(clear\ day|X=x)}$) is estimated as a linear function of the predictors for any given day:

$$log \frac{Pr(FLC\ day|X = x)}{Pr(clear\ day|X = x)} = \beta_0 + \beta^T x, \tag{1}$$

with $\beta_0$ the intercept and $\beta^T$ the model coefficients. From the odds ratio, the estimated probabilities and the corresponding class (FLC day or clear day) are determined for each sample. The ridge regression penalty based on the L2 norm, i.e. $R(\lambda) = \lambda \sum_{i=1}^{p} \beta_i^2$ is then incorporated as a constraint on the size of the regression coefficients in the objective function that is minimized to fit the model. The tuning parameter $\lambda$ directly trades off between a more flexible regression model (small

penalty, i.e., low $\lambda$ value) but that possibly suffers from high-variance issues, and a less flexible regression model. Accordingly, a larger value of $\lambda$ enforces smaller (but non-zero) regression coefficients, and a smoother spatial map of regression coefficients is obtained as a result. The optimal $\lambda$ value is derived through 10-fold cross validation. For a more complete description of regularized (logistic) regression, the reader is referred to Hastie et al. (2001), and the ElasticNet vignette for a hands-on tutorial (https://web.stanford.edu/~hastie/glmnet/glmnet_alpha.html). Model estimation and cross-validation was performed using the scikit-learn package in Python (Pedregosa et al., 2011).

The ridge regression method is used to predict FLC and clear days over the complete 14-year time series, using spatial patterns of 6 UTC (representative of averaging time of FLC cover, see Sec. 2.1) ERA5 MSLP fields in a large spatial domain centered on the central Namib (0°S–45°S and 8°W–38°E, shown in Fig. 2). The ERA5 pressure fields feature a spatial resolution of 0.25°x0.25° and as such, lead to 33, 485 predictor fields.

## 3 Results and discussion

### 3.1 Dynamic and thermodynamic conditions

Figure 2 shows a climatology of the dynamic and thermodynamic characteristics of the southeastern Atlantic and southern African region based on 14 years (2004–2017) of ERA5 data. Two seasons are shown in the figure that are representative of two different fog regimes (described in the next paragraph, Seely and Henschel, 1998; Andersen et al., 2019): September, October, November (SON) in the left-hand panels and April, May, June (AMJ) in the right-hand panels. At this spatial scale, the South Atlantic High and the continental high control the characteristic near-surface flow patterns during both seasons. During SON, the South Atlantic High is more prominent and, in combination with the thermal contrast between land and ocean, results in the formation of a low-level jet during this time (Nicholson, 2010). This alongshore coastal jet intensifies the upwelling of cold water, which feeds back to amplify the jet by increasing the thermal land-ocean contrast (Nicholson, 2010). On a local scale, the near-coastal winds that drive the upwelling are additionally modulated by the coastal topography (Koračin et al., 2004). While more prominent during SON, when the more pronounced South Atlantic High produces stronger winds, coastal upwelling water of the Benguela current is apparent in the relatively low SSTs along the southwestern African coastline during both seasons (Fig. 2 c) and d)), and throughout the year (Nelson and Hutchings, 1983). During AMJ, continental high pressure situations are the most prominent circulation pattern (Tyson et al., 1996; Garstang et al., 1996). This is visible in the more pronounced continental high pressure system and leads to a marked amplification of the easterly flow over the southern African continent. In the Namib Desert, thermally and topographically induced local wind systems within the boundary layer modulate these synoptic air-flow patterns, and the significance of the induced diurnal oscillations can exceed that of the synoptic scale (Goldreich and Tyson, 1988; Lindesay and Tyson, 1990). The combination of large-scale subsidence and low SSTs along the coastline produces high LTS conditions in the coastal marine regions adjacent to the Namib, specifically during SON (LTS contours in c) and d)). In the adjacent marine regions downwind of the central Namibian coast, these stable conditions promote the formation of the southeastern Atlantic stratocumulus cloud deck and controls its seasonal cycle (Klein and Hartmann, 1993; Andersen et al., 2017), where the SST component is responsible for most of the LTS seasonality. One should note that MSLP

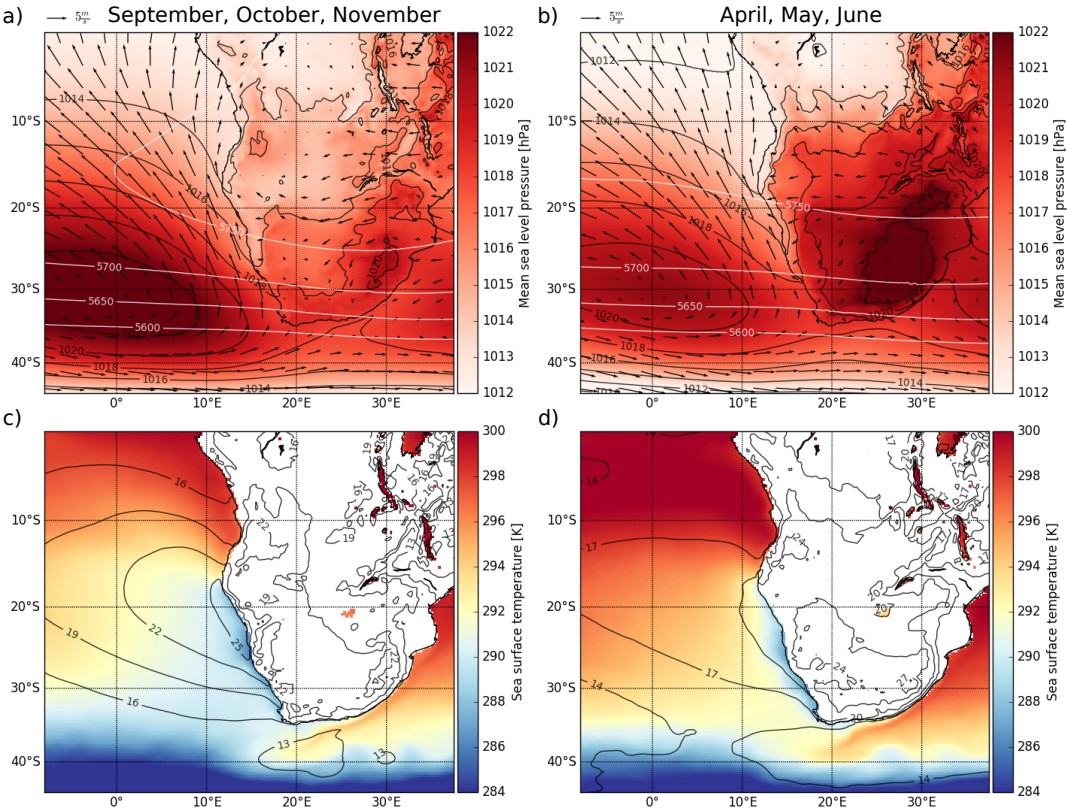

**Figure 2.** Climatological setting of the region in two seasons (2004–2017): September, October, November on the left and April, May, June on the right. Top row: MSLP in color and contours with 10 m winds indicated by arrows where the length scales with strength (the $u$ and $v$ vectors of near-surface winds are bilinearly interpolated to a 2.5°x2.5° grid for clarity). Z500 is illustrated with white contours. Bottom row: SST in color, LTS (K) as contours. Data is sampled at 6 UTC.

and LTS are both affected by the high elevation of the central plateau in southern Africa (cf. Fig. 6), and are not likely to be a perfect representation of near-surface pressure conditions and lower-tropospheric stability in this region. However, due to the joint consideration of regions in southern Africa with high topography, the low-lying central Namib, and marine regions, no one specific pressure level of geopotential height can adequately summarize near-surface conditions throughout this large domain. Additional analyses show that patterns obtained from MSLP fields in southern Africa are similar to those at 925 hPa and 850 hPa (not shown).

As outlined in the introduction, distinct seasonal fog and FLC patterns have been identified in the central Namib (Lancaster et al., 1984; Seely and Henschel, 1998; Cermak, 2012; Andersen et al., 2019). During SON, described as 'high FLC season' in Andersen et al. (2019), FLCs frequently occur in the central Namib as a low stratus or high fog (cloud base height on average $\approx$ 400 m above sea level (asl)) that touches the ground inland, whereas during the 'low FLC season' in AMJ, FLCs occur less frequently, do not extend as far inland and are typically lower, at $\approx$ 200 m asl and thus register as fog (termed 'advection

fog' in Seely and Henschel (1998)) at locations closer to the coastline (Andersen et al., 2019). While the FLC occurrence in the central Namib peaks in austral summer, and is lowest during winter, fog peaks at coastal locations in AMJ and at inland locations during SON (Seely and Henschel, 1998; Andersen et al., 2019) due to the seasonal cycle in the vertical position of the cloud layer (Andersen et al., 2019). For these reasons, this study focuses on mechanisms determining FLC variability within these two characteristic fog seasons.

It has been assumed that the occurrence of Namib-region FLCs and their variability on diurnal to seasonal scales is driven by the position and strength of large-scale pressure systems, as this would affect occurrence and advection of low-level clouds, atmospheric stability, and SSTs (Lancaster et al., 1984; Cermak, 2012; Andersen and Cermak, 2018; Andersen et al., 2019). Coastal upwelling, which has been shown to determine marine sea fog patterns along the Namibian coastline (Dorman et al., 2019), in combination with the presence of a coastal low that drives the onshore advection of foggy air masses have been found to be major drivers of fog occurrence in southern Namibia during austral summer (Olivier and Stockton, 1989). One should note though that the relationship between SSTs and Namib-region fog is complex, as Olivier and Stockton (1989) point out that a too large upwelling extent can also lead to less fog in southern Namibia. Based on these insights, and also on knowledge from related coastal upwelling systems (Cereceda et al., 2008; Johnstone and Dawson, 2010; Del Río et al., 2018; Dorman et al., 2019), it is clear that the Atlantic anticyclone, the SSTs, and the large-scale subsidence are main drivers of this coastal FLC system. While all of these links play a role for FLCs in the Namib, the influence of synoptic-scale variability has not been explored, and a more in-depth analysis is needed to estimate the importance of the different mechanisms for the day-to-day variability of Namib-region FLCs.

## 3.2 Differences in meteorological conditions on FLC days and clear days

Figure 3 shows large-scale patterns of averaged monthly mean differences in a) MSLP and 10 m winds, b) Z500 and winds at the same pressure level, c) T2m, d) LTS, e) SST, and f) TCWV on FLC versus clear days (as defined in Sec. 2.1) in the central Namib (marked with a star) during the investigated 14-year period (all months are considered here). The average of monthly mean differences is chosen rather than the overall mean differences to account for the distinct seasonal cycle of FLC occurrence in the Namib (Fig. 1 c)). In each pixel, an independent two-sided t-test is computed to identify significant differences between the two classes (contours show p values $<0.01$). It is apparent that the dynamical conditions (Fig. 3 a) and b)) on FLC days differ significantly on the synoptic scale. On FLC days, MSLP over continental southern Africa is systematically lower by about 3–5 hPa. This anomaly of lower MSLP extends over the southeastern Atlantic ocean at about 30°S. In a smaller oceanic region along the coastline north of 23°S, MSLP is significantly higher, leading to an overall anomalously high land-sea pressure gradient in this region and an onshore flow anomaly of near-surface winds in the central Namib on FLC days. The land-sea contrast in MSLP indicates a heat low over land, where the heat anomaly (Fig. 3 c)) could be driven by northerly advection ahead of the trough or enhanced surface warming. As discussed in Sec. 3.1, MSLP and 10m winds may not be a good representation of near-surface level characteristics where topography is high, however, additional analyses of geopotential height at 850 hPa and 925 hPa corroborate observed MSLP patterns. Differences exist in winds north of the central Namib, where at 925 hPa and 850 hPa (not shown), a stronger onshore flow anomaly is observed than at 10 m, possibly

indicating a topographical blocking of the onshore flow below the inversion. Z500 on FLC days (Fig. 3 b)) is significantly lower over the southeastern Atlantic between 30°S and 40°S. This pattern is an indication for upper-level waves disturbing the mean tropospheric circulation of the southeastern Atlantic and southern Africa (Tyson et al., 1996; Fuchs et al., 2017). In combination, MSLP and Z500 show a weakly baroclinic structure with the mid-level trough shifted to the west (cf. Fig. A1).

While a coastal low, which has been described in Olivier and Stockton (1989) as a local feature that can determine onshore flow, may still be present on FLC days, the composite differences between FLC days and clear days do not provide a clear indication of an increase in its presence on FLC days on average. However, as Reason and Jury (1990) describe, the coastal low is frequently followed by a frontal passage, which is a synoptic-scale signal observed here (Fig. A1).

There is a coherent pattern of slightly lower SSTs ($\approx 0.5$ K: Fig. 3 e)) along the coastline on FLC days; however, the
difference between SSTs on FLC and clear days is not significant at the 0.01 level (and also not at the 0.05 level). It is interesting to note that SSTs tend to be lower on FLC days, although the coast-parallel near-surface wind that partly governs the upwelling is slightly weaker in these cases (Fig. 3 a)), potentially hinting at a time-lag response of SSTs. This is to be expected, as Ekman transport produces a steady-state situation only after a few pendulum days (Pond and Pickard, 2013), although an initial upwelling response can be expected earlier (Lentz, 1992). It appears likely that effects of SST patterns on
FLC variability are most pronounced on longer time scales (i.e. seasonal to interannual) that feature higher SST variability (Hutchings et al., 2009; Goubanova et al., 2013; Tim et al., 2015), as also observed in the Chilean Atacama desert (Del Río et al., 2018). Differences in TCWV on FLC and clear days are pronounced (Fig. 3 f)). A coherent region of a significantly dryer column stretches from the central Namib over the coastal Atlantic, where the anomaly is strongest. This is likely the dry slot (Browning, 1997) or dry air intrusion of the synoptic-scale disturbance that leads to increased longwave cooling at cloud
top in case of FLC presence, which has been shown to be a main determinant of cooling within the marine boundary layer (Koračin et al., 2005). This enhanced cooling can increase FLC cover, which has been observed to be a significant mechanism for stratocumulus clouds over the southeastern Atlantic (Adebiyi et al., 2015; Adebiyi and Zuidema, 2018). A substantial moist anomaly is visible over the southern African continent, likely driven by large-scale free-tropospheric moisture transport from the north west (Fig. 3 b)). These moist air masses may contribute to the observed T2m heat anomaly via greenhouse warming
(Fig. 3 c)). This effect of free-tropospheric moisture on surface temperatures has been observed in the Kalahari (Manatsa and Reason, 2017) and other arid or semi-arid regions before (Evan et al., 2015; Oueslati et al., 2017; Alamirew et al., 2018). Along the coastal strip that is typically overcast with FLCs (Olivier, 1995; Cermak, 2012; Andersen and Cermak, 2018; Andersen et al., 2019), T2m is significantly lower by about 4 K, which is likely a feedback of FLCs reflecting solar radiation and slowing down the surface heating in the early morning (Iacobellis and Cayan, 2013), or due to air-mass differences. The observed
difference patterns in LTS (Fig. 3 d)) between FLC and clear days matches those of T2m so that they can be assumed to be mostly driven by its surface component (Pearson correlation coefficient is -0.90 for land pixels).

The observed anomaly patterns indicate that different mechanisms are triggered by the observed synoptic-scale disturbances and may contribute to FLC occurrence in the central Namib in two main ways:

1. Increased FLC cover due to increased longwave cooling under the dry anomaly close to the coast.

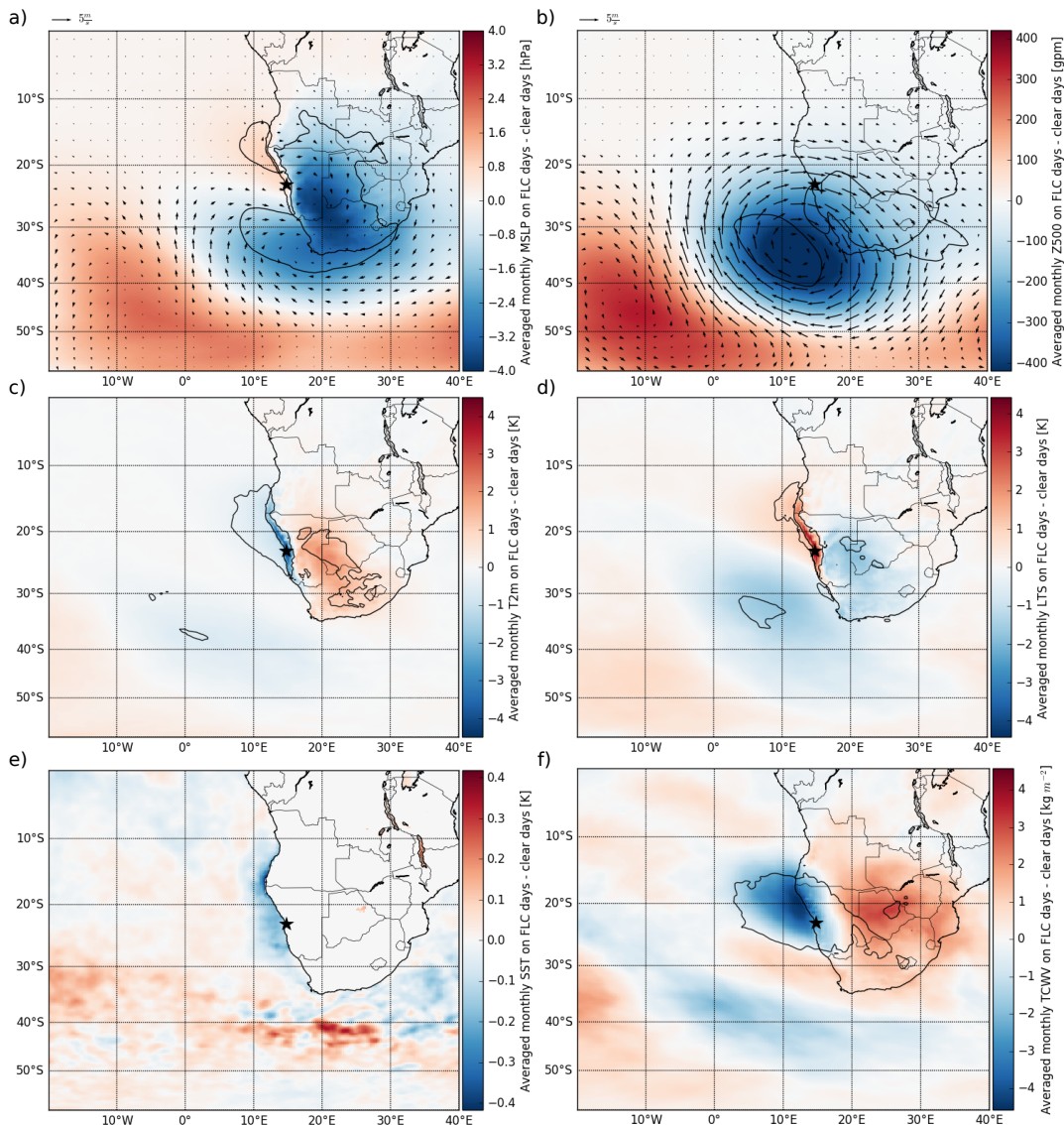

**Figure 3.** Averaged monthly mean differences (FLC days - clear days) of a) MSLP and 10 m winds, b) Z500 and 500 hPa winds, c) T2m, d) LTS, e) SST, and f) TCWV at 6 UTC. In each pixel, an independent two-sided t-test is computed to identify significant differences between FLC and clear days for each month. Contours mark regions where the distributions differ significantly at the 0.01 level (median of the monthly p values < 0.01). $U$ and $v$ vectors of winds are interpolated as in Fig. 2.

2. Onshore flow anomaly of marine boundary layer air masses due to a) a modulation of coastal winds and b) a formation of a southern African heat low due to greenhouse warming by moist air masses and northerly warm air advection.

As both synoptic and FLC characteristics differ substantially between the SON and AMJ, the following section focuses on specific characteristics and differences of these mechanisms during these seasons.

## 3.3 Seasonal differences in synoptic-scale mechanisms

Figures 4 and 5 show seasonally averaged differences between FLC and clear days of all analyzed parameters during the two seasons SON and AMJ. During both seasons, MSLP (Fig. 4 a) and b)) and Z500 (Fig. 4 c) and d)) indicate synoptic-scale disturbances on FLC days. However, seasonal differences exist, as the disturbance is more pronounced during AMJ. The negative continental MSLP anomalies on FLC days are larger during AMJ, likely amplified by the more pronounced T2m anomalies and subsequent effects on a continental heat low during this time (cf. Fig. 5 c) and d)). As noted above, the continental heat anomaly can be caused by northerly warm air advection or enhanced warming due to changes in the radiative balance. The observed seasonal MSLP and 10 m wind anomalies (Fig. 4 a) and b)) that result in a transport of warm air from the northeast into the anomaly region (see Fig. A2), as well as the TCWV anomalies (Fig. 5 a) and b)) suggest that during SON, the heat anomaly on FLC days is mostly due to northerly advection of warm air, whereas during AMJ, TCWV is significantly increased over the southern African continent. Here, the T2m anomalies closely follow those of TCWV (Pearson correlation coefficient of 0.75 in continental regions with significantly higher T2m on FLC days than clear days), suggesting that the increased moisture causes an additional surface heating due to greenhouse warming as discussed in Sec. 3.2. It is likely that the TCWV anomaly is caused by a large-scale free-tropospheric moisture transport from the tropics, which is supported by the marked wind anomalies at 500hPa (Fig. 4 d)) that show a northwesterly anomaly, and the absolute wind and moisture fields at 700 hPa during this time (Fig. A1). It should be noted that a Lagrangian transport of moisture at this scale takes time and as such is likely to occur when the disturbance is relatively stationary or if two consecutive systems pass within a short timeframe (Knippertz and Martin, 2005).

While the yearly averaged composites show that over land, LTS is driven to a large extent by T2m (Fig. 3 c) and d)), this is not quite as pronounced during SON (correlation coefficient =-0.57; Fig. 5 c) and e)). Over continental southern Africa, the differences in T2m (Fig. 5 c)) are frequently compensated by similar differences in potential temperature at 700 hPa (not shown). The most pronounced LTS feature during both seasons, however, is the coastal anomaly of increased LTS (over land and weaker over the adjacent ocean), which is driven by T2m. As this anomaly is also apparent during nighttime (1 and 3 UTC, not shown), it is likely that this pattern is mainly due to the relatively warm subsiding continental outflow that is apparent on clear days, rather than a radiative effect of FLCs as found in California (Iacobellis and Cayan, 2013). During AMJ, LTS is significantly lower over a large marine region south of 25°S, which is likely caused by the synoptic-scale disturbance.

During both seasons, SSTs in the coastal upwelling region are slightly lower on FLC days than on clear days, although these differences are not significant at the 0.01 level for the most part (very localized regions at ≈28°S are significantly lower during AMJ). In isolated patches further south, upwind of the study area, SSTs tend to be significantly higher on FLC days. This could lead to increased surface latent heat fluxes, increasing the moisture content of the marine boundary layer, particularly during AMJ when stronger near-surface winds are also apparent. A few 100 km to the west and south of the Namibian coastline, SSTs could similarly add to the increased moisture within the marine boundary layer. It is not clear yet, however, what exactly

drives the observed anomaly patterns of SSTs. As upwelling reacts to the time-integrated wind field forcing over longer time scales than analyzed here (Pond and Pickard, 2013), the SST response to the instantaneous winds that are considered here is expected to be relatively weak. However, in the case of a relatively stationary disturbance as discussed above, the upwelling patterns could indeed reflect an SST response to a synoptic forcing. While the seasonally varying TCWV and SST anomalies

(Figs. 4 e) and f), and 5 a) and b), respectively) illustrate the seasonal variability in the mechanisms that can contribute to FLC occurrence in the central Namib, during all months, the outlined systematic patterns of significant negative MSLP anomalies over continental southern Africa and the localized coastal high pressure anomaly are apparent. It can be concluded that a low pressure anomaly in continental southern Africa and the associated onshore advection of marine boundary layer air masses facilitates FLC occurrence in the central Namib during the entire year.

To better understand the characteristics of the observed moisture transport and its relevance for central-Namib FLC occurrence, information on the vertical patterns of moisture and wind anomalies is needed. Figure 6 shows average seasonal differences of Q and winds on FLC versus clear days at different pressure levels during a) SON and b) AMJ (averaged between 20°S and 25°S). During both seasons, a complex vertical structure of Q anomalies is apparent that is assumed to be disturbance-induced. During both seasons, the marine boundary layer features an onshore flow anomaly and is more humid

on FLC than on clear days, especially during AMJ, where this is a synoptic-scale feature, likely related to the cold front of the disturbance. These differences are caused by the subsiding dry continental easterly air masses that dominate on clear days, whereas on FLC days, a slight onshore flow of the more humid marine boundary layer air is observed in the central Namib. Over land, these marine air masses flow against the dominant continental easterly winds (Lindesay and Tyson, 1990), producing a northerly wind flow at $\approx 15°$N (not shown) that has been found to be associated with fog occurrence in the central Namib

(Seely and Henschel, 1998; Spirig et al., 2019). Above the moist marine boundary layer, the free troposphere is relatively dry on FLC days during both seasons, a feature which is not as clearly visible in the columnar TCWV composites during AMJ as it is masked by the moist anomaly in the marine boundary layer (Fig. 5 a)). It is interesting to note that the marine dry anomaly peaks between December and February (not shown), the season with maximum FLC cover in the central Namib, with TCWV anomalies exceeding 10 kg m$^{-2}$. The seasonal difference in the free-tropospheric Q anomalies over the continent is

clear and the vertical distribution of Q anomalies during AMJ corroborates the assumption that the observed positive TCWV anomalies are due to free-tropospheric moisture transport (Fig. 5 b)). Expressed in relative terms, Q is about halved within the dry anomaly region on FLC days during both seasons, suggesting that radiative cooling is an important factor for FLC cover, especially over marine regions where the dry anomaly is most pronounced. During AMJ, the free-tropospheric relative moisture difference between FLC days and clear days is observed to be as high as 220 %. This substantial increase in free-tropospheric

moisture in this otherwise dry central plateau region induces a substantial surface heating, contributing to the formation of the observed heat low, which modulates regional wind systems and leads to the onshore flow anomaly.

It should be noted that in a comparable upwelling system (coastal California), Clemesha et al. (2017) also find a positive relationship between T2m over land and coastal low-level cloudiness, with the T2m anomaly shifted poleward by about 5° latitude from the cloud field. They propose that the T2m-cloud relationship is due to spatially-offset associations between

coastal low-level cloudiness and stability (potential temperature at 700 hPa), which is strongly correlated to T2m over land

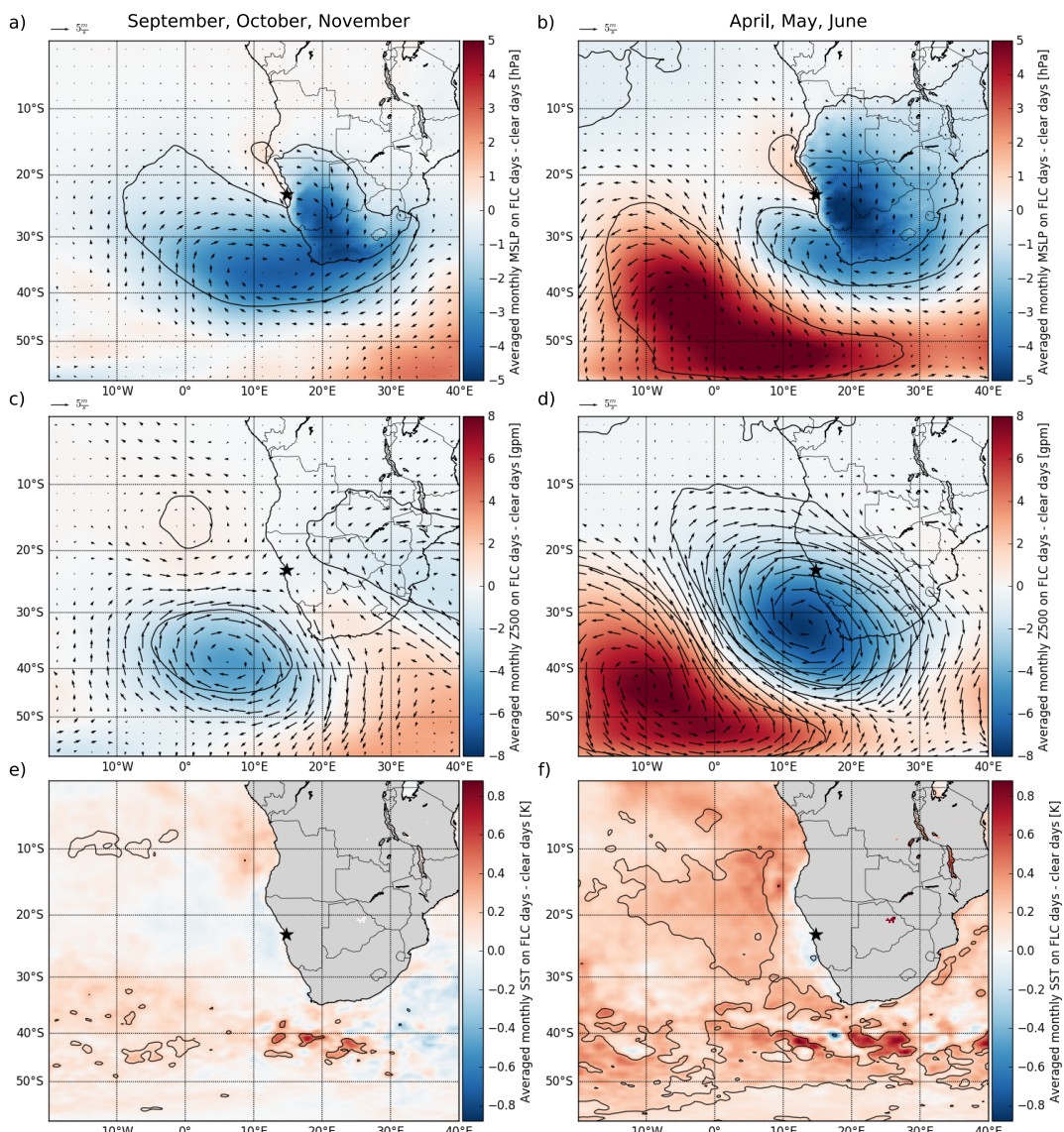

**Figure 4.** Mean of monthly average differences (FLC days - clear days) during SON (left-hand panels) and AMJ (right-hand panels) of MSLP (top), Z500 (middle), and SST (bottom) for the time period 2004–2017. Contours mark significant differences as in Fig. 3. Wind anomalies at 10 m (top) and 500 hPa (middle) are superimposed as vectors.

thereby resulting in the T2m anomaly, rather than T2m driving the onshore advection. While in the central Namib, the anomaly patterns between potential temperature at 700 hPa and T2m are similar in that they are also positively correlated during SON (and therefore compensate each other in terms of LTS, Fig. 5 c) and e)), they are uncorrelated during AMJ (and also in the annual averages), when T2m over land is strongly correlated to TCWV. Also, during all times of year, the T2m and MSLP anomalies are directly inland from the cloud field, suggesting an influence on onshore advection.

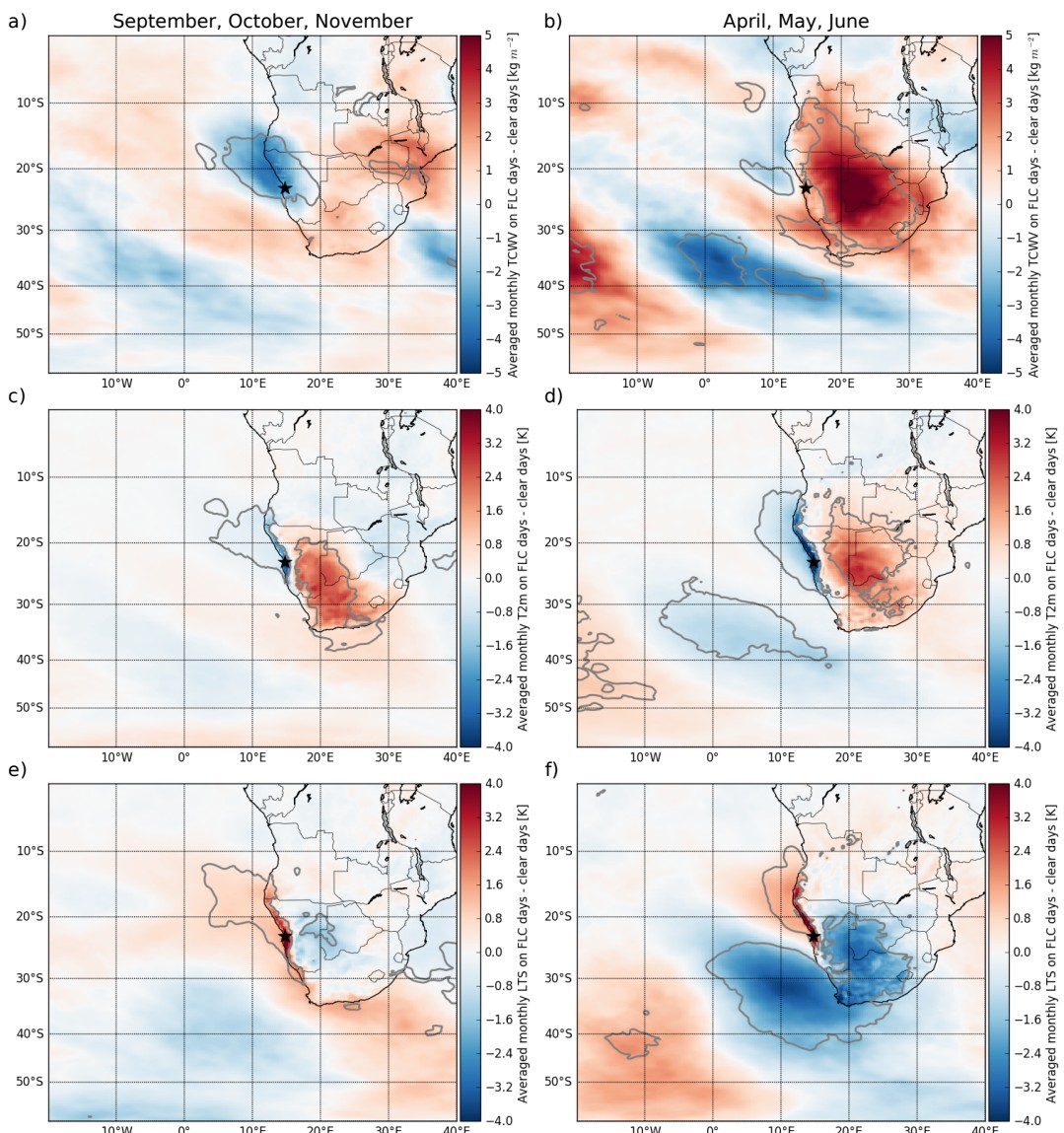

**Figure 5.** Mean of monthly average differences (FLC days - clear days) during SON (left-hand panels) and AMJ (right-hand panels) of TCWV (top), T2m (middle), and LTS (bottom) for the time period 2004–2017. Contours mark significant differences as in Fig. 3.

## 3.4 The role of air-mass history and dynamical regimes

Air-mass backtrajectories, initiated in the central Namib close to Gobabeb at 23°S and 15°E (indicated by the star in Fig. 7), 6 UTC and 25 hPa above ground level (approximates 200 m above ground level), are computed for the 14-year observational period. Figure 7 shows the backtrajectories for FLC days (top) and clear days (bottom) for the two seasons SON (left-hand panels) and AMJ (right-hand panels). During both seasons, air masses on FLC days nearly exclusively stem from the marine

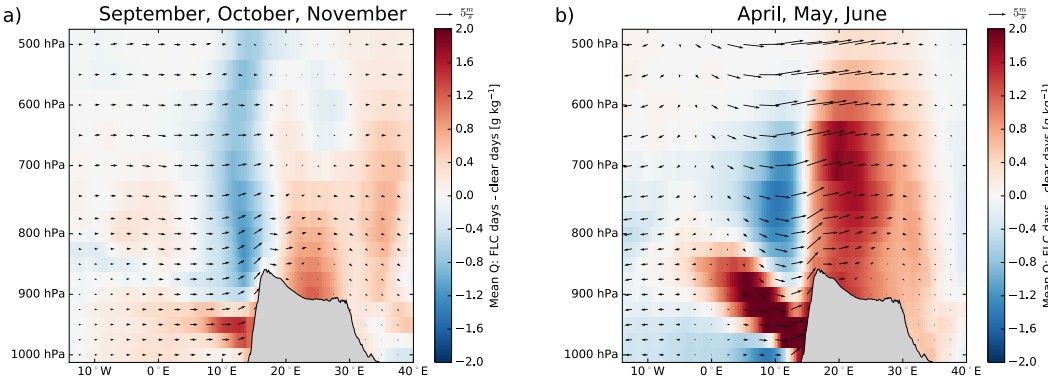

**Figure 6.** Seasonal average difference (FLC days - clear days) in specific humidity, and u and w wind components at different pressure levels during a) SON and b) AMJ for the time period 2004–2017. Specific humidity and wind vectors are averaged between 20°S and 25°S and shown at pressure levels between 1000 hPa and 500 hPa. For illustration purposes, the w vector is enhanced by a factor of 20. The masked grey area approximates the average surface elevation between 20°S and 25°S.

boundary layer and have traversed over the cool upwelling ocean water along the coastline for the time span of 24 h. This is in agreement with findings from Koračin et al. (2005), who note that a marine origin of air masses is critical, as well as potential mixing with continental air masses along the trajectory that would lead to a warming and drying. While the number of FLC days during SON is higher than during AMJ, following the general seasonality of FLCs in the region (cf. Fig. 1 c)), no clear

seasonal differences in air-mass dynamics can be observed in such situations. This suggests that during both seasons, similar local dynamic conditions drive FLCs or air masses that develop into FLCs inland into the Namib desert, but that due to seasonal differences of large-scale dynamics, these situations occur with varying frequency during different seasons.

On clear days, air-mass histories are more diverse and show distinct seasonal differences, but are frequently characterized by subsiding continental air masses. While on clear days during SON, a considerable fraction of the air masses is still transported

from the marine boundary layer, during AMJ, subsiding north-easterly continental air masses dominate. This seasonal shift in air-mass dynamics is likely driven by the seasonality of the two dominating high pressure systems of the region that is shown in Fig. 2. During AMJ, the continental high pressure system is enhanced and leads to the stronger easterly flow. These observations support the hypothesis by Lancaster et al. (1984) that the seasonality of fog in the central Namib is to some extent controlled by the Southern African high pressure system, as the associated easterly winds are likely to inhibit large-scale

onshore advection of cloudy marine boundary layer air masses. The results also suggest that aerosols from the biomass burning season in continental southern Africa (Swap et al., 2003) are unlikely to play a large role for fog formation by acting as cloud condensation nuclei, as biomass burning aerosols within the boundary layer are mostly associated with continental air masses in this region (Formenti et al., 2018). However, biomass burning aerosols may influence Namib-region FLCs by absorbing solar radiation and modifying the thermodynamic conditions, which has been observed and modeled to influence the Namibian

stratocumulus deck (Zhou and Penner, 2017; Deaconu et al., 2019).

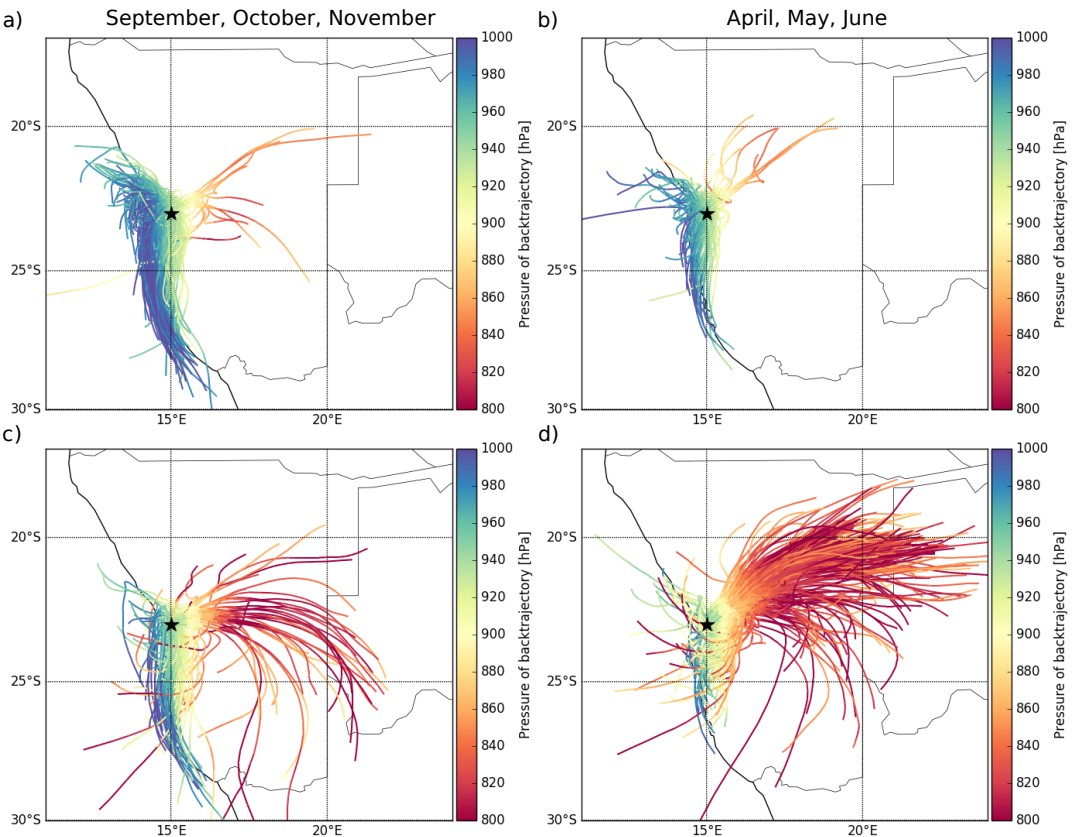

**Figure 7.** 24-hour Lagranto air-mass backtrajectories for FLC days (top) and clear days (bottom) in September, October, November (left-hand panels) and April, May, June (right-hand panels) for 2004–2017. The star marks 23°S and 15°E, where the backtrajectories were initialized at 25 hPa (approximates 200 m) above ground level. The number of samples are a) 399, b) 133, c) 146 and d) 452.

While systematic differences in air masses exist between FLC days and clear days, clear days may still feature air masses that are advected from the marine boundary layer (cf. Fig.7 c)). To understand the differences between the FLC days and clear days in such situations, these are isolated and analyzed in the following. Figure 8 shows the average Q, relative humidity (RH), air temperature (T), potential temperature (Pot. T), and pressure (P) along all of the backtrajectories that are advected from the marine boundary layer (here: P > 900 hPa over ocean). It is apparent that these air masses contain significantly more moisture and feature significantly lower Pot. T on FLC days than on clear days, which explains most of the difference in RH. The backtrajectories of FLC days feature a stronger cooling during the last 10 hours of advection (hours 0–10), resulting in an additional increase in RH. The deviation in T between FLC and clear days seems to be driven by the vertical movement of the air masses, rather then differences in radiative cooling, as no changes in Pot. T are apparent. Ten hours before initialization, air masses on clear days are located ≈20 hPa higher than on FLC days, not cooling off as they are advected due to their simultaneous subsidence. Other potential factors that may drive the observed deviation in T, such as the free tropospheric

moisture content and the surface temperature along the backtrajectories, were not found to be systematically different on FLC and clear days (not shown). These findings highlight that Namib-region FLCs are not only dependent on dynamics, but that marine boundary-layer moisture content as well as temperature changes during advection are important controls as well.

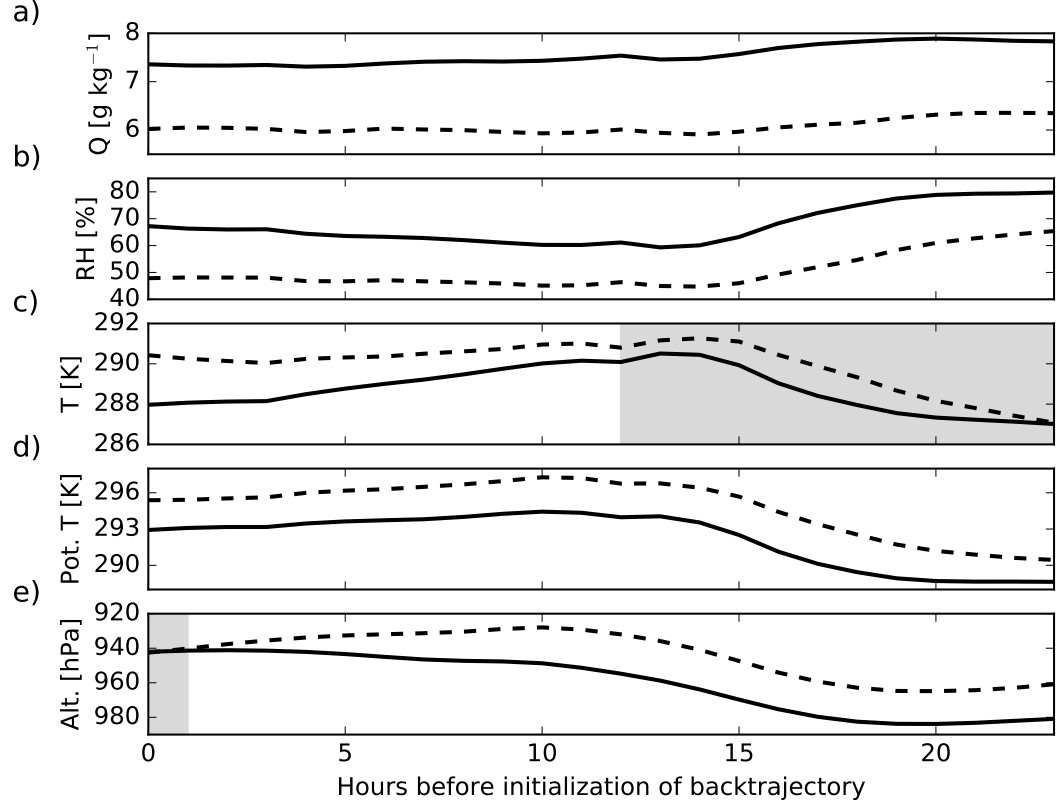

**Figure 8.** Hourly averaged specific humidity (a)), relative humidity (RH, b)), air temperature (T, c)), potential temperature (Pot. T, d)), and pressure (P, e)) along the 24-hour air-mass backtrajectories that are advected from the marine boundary layer on FLC days (solid line) and clear days (dashed line) during SON 2004–2017. The number of samples are 369 FLC days and 80 clear days. Grey shading highlights non-significant differences.

It is likely that the computed air-mass backtrajectories do not fully capture thermally and topographically induced local air flow patterns (see Lindesay and Tyson (1990) for a review) that contribute to local FLC occurrence patterns and possibly formation. However, the larger-scale patterns of air-mass history of marine boundary-layer air masses versus the subsiding continental air masses from the free troposphere are clearly evident from the analysis presented and offer a consistent physical explanation of the large-scale FLC occurrence patterns. The observations suggest that Namib-region FLCs are either advected after forming over the cool adjacent ocean or that condensation takes place during advection of the marine boundary-layer air masses over land due to higher humidity levels, lower temperatures or radiative cooling, though a mix of these processes is likely.

The analysis of air-mass backtrajectories shows that the discrimination between FLC and clear days is not possible using dynamics alone, and that seasonal differences exist in the link between the probability of FLC days and advection patterns. To further investigate the role of different dynamical regimes for FLC occurrence, a PCA is conducted on spatial patterns of synoptic-scale near-surface winds (see Sec. 2.4 for details on the method). Figure 9 a) shows correlations between daily central-Namib FLC cover and the PCs associated with the first six modes of variability of near-surface winds during all months of the year. All PCs are significantly correlated to FLC cover during some months of the year. Clear correlation patterns are evident: PCs 1, 2, 4 and 5 show negative correlations with FLC cover, while PCs 3 and 6 feature positive correlations. These PCs that facilitate FLC occurrence (3, 6) all show westerly or northwesterly wind anomalies in the central Namib, while PCs that are negatively associated with Namib-region FLC cover feature anomalously strong continental easterly winds, consistent with results presented in Sec. 3.2 and 3.3. Panels c) and d) of Fig. 9 show the spatial patterns of near-surface wind anomalies of PCs 3 (explained variance: 11 %) and 4 (explained variance: 6 %), as examples for PCs that promote and impede FLC occurrence, respectively. A seasonal dependence of the correlations between PCs and FLC cover is apparent and seems to be related to the seasonality of FLC cover (Fig. 9 b)): PCs associated with onshore circulation in the central Namib feature the strongest positive correlations during winter when FLC cover is generally lowest over the Namib, especially evident for PC 3. This appears plausible, as during winter, the typical dynamical setting is less conducive to FLCs (see Fig. 2 for fall/early winter conditions during AMJ), and consequently, FLC occurrence is dependent on a stronger dynamical disruption during this time. During summer, when FLCs frequently occur in the central Namib, dynamical conditions associated with PCs 4 and 5 (dominance of continental easterlies) seem to impede the occurrence of FLCs. The results underscore that the advection of marine air masses is crucial for the occurrence of FLCs in the central Namib.

## 3.5   Statistical fog and low-cloud prediction with pressure fields

Based on the evidence presented above, showing that FLC occurrence is tightly connected to synoptic-scale patterns, it can be assumed that FLC occurrence can be predicted to some extent with a statistical learning technique that utilizes spatial patterns of dynamical information. Here, a ridge regression is applied to classify FLC days and clear days based on MSLP fields in a region spanning 45°x45° that is centered on the central Namib (see Sec. 2.4). MSLP fields are used, as their anomaly patterns on FLC days are similar during the different analyzed seasons and thus summarize the controlling mechanisms of onshore advection of marine boundary-layer air masses. Figure 10 a) shows the resulting coefficients, i.e. regression slopes, of the statistical model. The sign and spatial patterns of the coefficients are similar to the observed MSLP anomalies shown in Fig. 4, where coefficients (and anomalies) are negative in the inland region of Namibia, and positive and along the northern part of the Namibian coastline. It should be noted that the statistical model seems to mostly rely on regional MSLP fields, resulting in low coefficients at the synoptic scale, e.g., the Atlantic high pressure system. It can be concluded that the synoptic-scale pressure patterns set the stage for more localized pressure and wind modulations that determine FLC occurrence, and that regional MSLP fields contain information on both.

Figure 10 b) gives a summary of statistical measures of the skill of the model to classify between FLC and clear days in the central Namib. Using MSLP fields at 6 UTC on the day of the FLC cover information, the ridge regression model has a

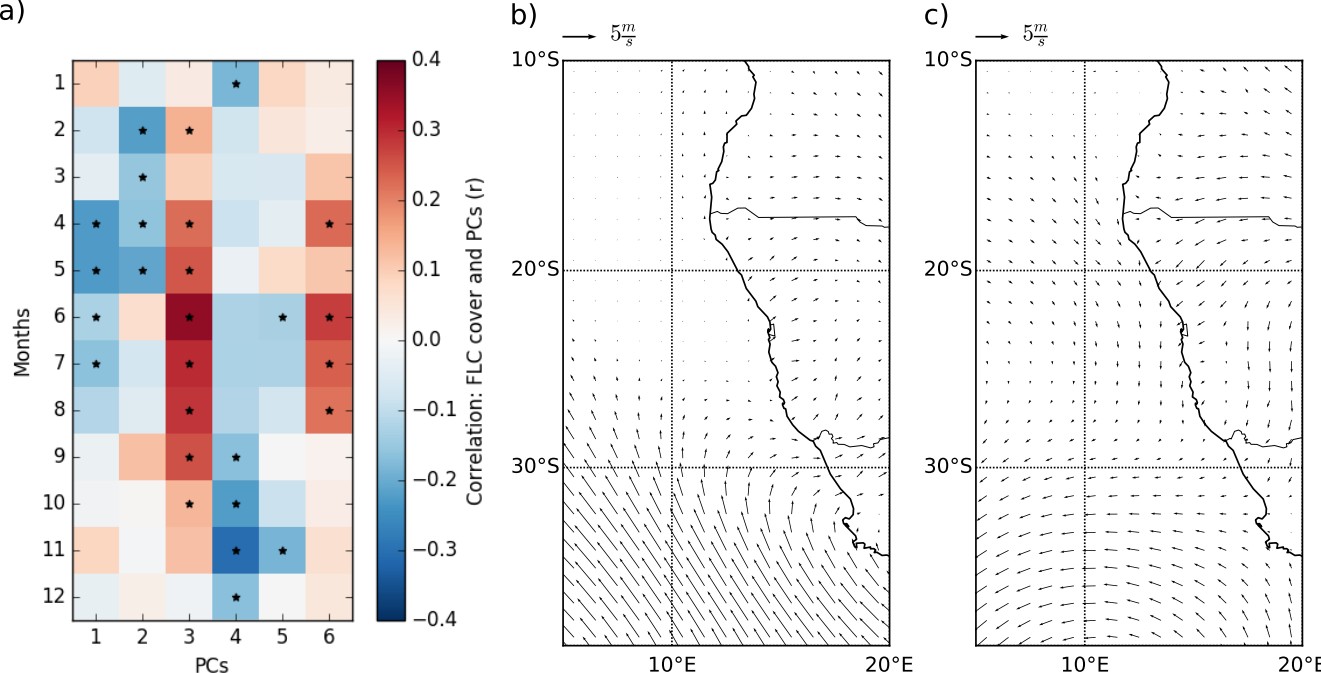

**Figure 9.** a) Correlations (Pearson r) between the PCs (associated with the empirical orthogonal functions of the spatial wind patterns) and central-Namib FLC cover. Stars mark correlations that are significant at the 0.01 level. Panels b) and c) show wind anomaly fields for PCs 3 and 4, respectively. For visual clarity, spatial wind anomalies are shown for regional cutouts of the spatial domain that is considered in the PCA and averaged to a 1° x 1° resolution (see Sec. 2.4 for details on the analysis).

probability to correctly detect FLC days of 94 % with 17 % of the reported FLC days being false alarms, leading to an overall correctness of the model of 86 % and a positive bias of 14 %. The critical success index (CSI: 0.79), and the Heidke skill score (HSS: 0.72) combine these scores and show that the model is skillful in distinguishing between the defined fog and clear days. As MSLP fields in southern Africa may not be representative due to the high topography, the model was additionally run based on Z850 and Z925 fields. The model performances were nearly the same (overall PC of 84 % in both cases), suggesting that it is adequate to use MSLP in this context. The colored dots in Fig. 10 b) illustrate the progression of the model skill when the training is carried out based on MSLP fields of one, two, three or four days prior to the FLC observation. While, as expected, the model skill deteriorates with an increasing temporal gap between the MSLP predictors and the time of FLC occurrence, the model is capable of predicting fog occurrence fairly well one day in advance, as the time series of day-to-day FLC occurrence features a significant autocorrelation of some days. To some extent, this may be connected to the strong persistence of synoptic-scale dynamics in the subtropics. Even though the model only uses MSLP fields, ignoring e.g. effects of radiative cooling due to moisture anomalies, surface temperatures, and seasonal characteristics, which have been shown to modify FLC occurrence, the results still illustrate the potential of a dynamics-based statistical fog forecast in this region. It should be noted that changes in circulation additionally influence upwelling intensity (e.g. Hutchings et al., 2009) such that some of the explained variability

may also be attributed to factors influencing FLC formation rather than advection. However, due to the longer time-scale of SST responses, and due to the marked contrasting differences in air mass history on FLC and clear days, the latter is thought to be the first order mechanism leading to the high model skill.

It should be noted that the distinction between FLC and clear days is based on spatially and temporally averaged FLC occurrence (see Sec. 2.1) and that days are omitted that feature an FLC cover between 3 % and 50 %. Also, the exact location and time of FLC occurrence is likely to be dependent on local temperature gradients and topography that lead to local modulation of winds (Lindesay and Tyson, 1990). Still, the model produces promising results that may be built upon in future studies by testing a similar model setup to predict the timing and duration of FLCs at specific locations.

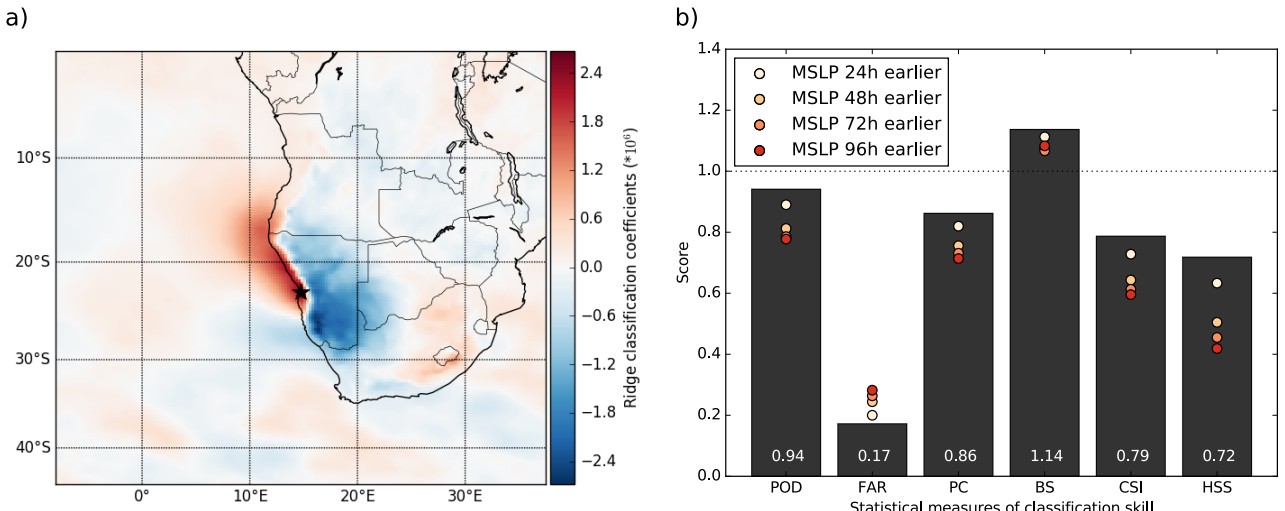

**Figure 10.** a) The coefficients of the ridge regression used for classification of FLC days versus clear days. b) Statistical measures of the performance of the ridge regression to classify FLC days versus clear days. The bars and related numbers describe the model skill using 6 UTC MSLP fields of the day of FLC observation and relate to a). The colored dots show the model skill when the model is trained on MSLP fields of one to four days earlier. The abbreviations of the statistical measures stand for probability of detection (POD), false alarm rate (FAR), percent correct (PC), bias score (BS), critical success index (CSI), and the Heidke skill score (HSS). The equations of the statistical measures are given in the appendix.

## 4  Summary and conclusions

In this study, the occurrence of FLCs in the Namib desert, derived from 14 years of satellite observations, is systematically analyzed within the context of the regional climate and related to large-scale patterns of MSLP, Z500, T2m, LTS, SST, TCWV as well as Q and winds at different pressure levels from ERA5 reanalyses. The satellite data set of FLC occurrence is separated into FLC days and clear days that are further investigated in terms of their meteorological conditions, air-mass histories and statistical predictability during two seasons (AMJ and SON).

It is found that MSLP and Z500 patterns on FLC days are systematically and significantly different from clear days on synoptic scales. On FLC days, a systematic pattern of significantly lower MSLP over continental southern Africa is observed, which, in combination with higher pressure over a marine coastal region at about 20°S, leads to an onshore flow anomaly of marine boundary-layer air. Together with significantly lower Z500 in the southeastern Atlantic region on FLC days, these dynamic patterns are an indication for synoptical-scale disturbances. These modify circulation systems, which in turn alter moisture transport, resulting in characteristic moisture patterns on FLC and clear days. Over the coastal boundary layer, the free troposphere is observed to be significantly drier on FLC days during both seasons, increasing radiative cooling, which likely increases FLC coverage, especially over the ocean where the dry anomaly is observed to be most pronounced. During AMJ, free-tropospheric moisture over the southern African continent is substantially increased, leading to greenhouse warming at the surface. While northerly warm air advection also contributes to the observed positive T2m anomalies on FLC days (during both seasons), the additional increase in T2m on FLC days during AMJ clearly corresponds to regions of increased free-tropospheric moisture content (correlation = 0.75). The increase in T2m leads to the development of a heat low that amplifies the upper-level disturbance-induced low MSLP anomaly, thereby contributing to the onshore flow anomaly of marine boundary-layer air masses. In the localized coastal region where FLCs typically occur, T2m is found to be significantly lower on FLC days, likely a combination of a local feedback of FLCs that slow down surface heating in the morning hours, and air mass differences. A significant pattern of SST anomalies is found only in AMJ, with anomalously high SSTs off the coast possibly acting together with increased near-surface winds to enhance surface latent heat fluxes that may contribute to the observed higher levels of specific humidity in the marine boundary-layer.

The analysis of backtrajectories initialized in the central Namib at typical cloud level shows systematic differences in air-mass dynamics on FLC days and clear days. Air masses on FLC days are nearly exclusively transported within the marine boundary layer over the cool upwelling waters along the coastline, whereas clear days are frequently associated with subsiding northeasterly air masses, especially during AMJ. During SON, when advection of marine-boundary layer air masses can also occur on clear days, air masses on clear days feature significantly less moisture and tend to be advected from higher altitudes than on FLC days. The findings clearly demonstrate the strong dependence of central-Namib FLC occurrence on the advection of moist marine boundary-layer air masses, contrasting the notion of predominant radiation fog (Kaseke et al., 2017), but in agreement with many other studies (e.g. Olivier and Stockton, 1989; Seely and Henschel, 1998; Formenti et al., 2018; Andersen et al., 2019; Spirig et al., 2019). These results are supported by a principal component analysis of near-surface winds that show a clear connection of FLC cover to synoptic-scale dynamics. Principal components of spatial wind patterns that feature positive onshore flow anomalies are positively related to FLC cover. This relationship is especially strong during winter, when FLC occurrence is at its minimum, as then, the dominant continental easterly flow typically inhibits inland advection of FLCs or locally developing FLCs. This suggests that during this time, a stronger dynamical forcing is needed to overcome this characteristic flow that is unfavorable for inland advection of cloudy marine boundary layer air masses.

As the results show that spatial pressure patterns are connected to FLC occurrence, a ridge regression model is used to classify FLC days versus clear days based on regional MSLP fields. The resulting spatial pattern of model coefficients is similar to the observed MSLP anomaly patterns within the region of Namibia and the adjacent ocean areas. The spatial domain

of relevant model coefficients seems to be smaller than the spatial extent of the pressure anomalies, probably because the regional fields contain information on synoptic-scale disturbance as well as local modulation. On this basis, the model is capable of skillfully delineating FLC days from clear days. The model is trained with MSLP fields with different temporal offsets, and found to be capable of skillfully predicting FLC occurrence one day in advance, highlighting the potential of a statistical forecast of FLCs in this region. Future work should focus, however, on the development of a statistical model that links information on e.g. MSLP, free-tropospheric moisture, SSTs, Z500 and aerosol loading with FLC occurrence to quantify the effects of the different processes and mechanisms outlined in this study.

The findings of this study suggest that FLCs in the central Namib are facilitated by synoptic-scale disturbances in two main ways:

1. Increased longwave cooling due to an anomalously dry free troposphere, especially over the ocean that increases low-cloud cover.

2. Onshore flow anomaly of these cloudy marine boundary layer air masses due to

   a) disturbance-induced modulation of local winds, and

   b) a heat low over continental southern Africa.

The magnitude and characteristics of the disturbance and the related mechanisms depend on season, with a more pronounced disturbance during AMJ, when the typical dynamic setting is less conducive to FLC occurrence. Figure 11 is a schematic illustration that summarizes these seasonally varying mechanisms.

While a 14-year sample is not optimal to capture climatological variability, the mechanisms documented here for the first time are unlikely to be fundamentally different in other climatological periods. While it seems settled that, at least at the scales considered in this study, FLC occurrence is mostly driven by advective processes, the quantitative contributions of humidity and temperature changes and radiative cooling for low-cloud formation in the Namib during the advection of marine boundary-layer air masses are still unclear. A heat budget analysis as in e.g., Adler et al. (2019) or Babić et al. (2019), based on ground-based measurements conducted during the field campaign of the Namib Life Cycle Analysis (NaFoLiCA) project (Spirig et al., 2019), is necessary to better understand the origin, development and life cycle of FLCs within the advected marine boundary-layer air masses. Future work should also focus on understanding the local and possibly synoptic-scale drivers of the vertical structure of Namib-region FLCs on diurnal to seasonal scales, and the day-to-day variability of (marine) boundary-layer humidity. As FLCs in the Namib are clearly connected to marine stratus/stratocumulus clouds, findings of recent and ongoing field campaigns over the southeastern Atlantic (Zuidema et al., 2016; Formenti et al., 2019) and related insights concerning the aerosol-cloud-meteorology system of the Namibian stratocumulus cloud field (e.g., Adebiyi and Zuidema, 2018; Andersen and Cermak, 2015; Diamond et al., 2018; Formenti et al., 2019; Fuchs et al., 2017, 2018; Gordon et al., 2018) are relevant to fully understand FLCs in the Namib desert.

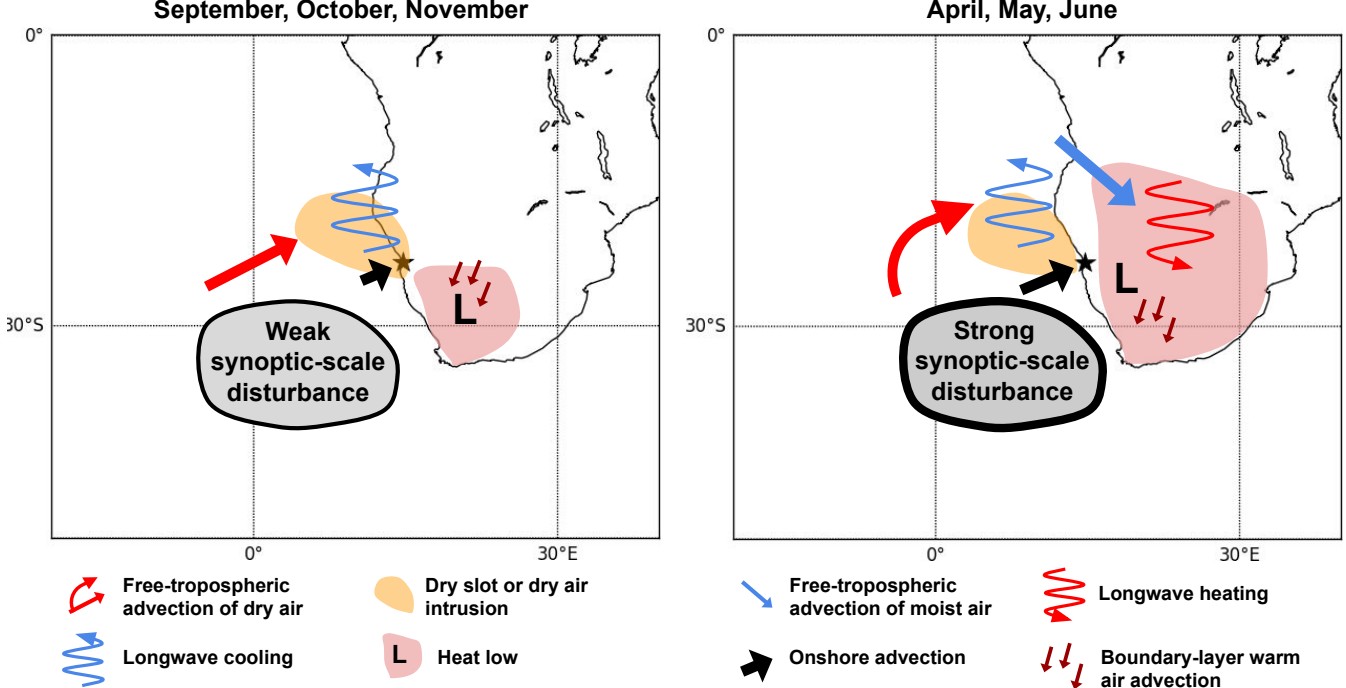

**Figure 11.** Schematic overview over the synoptical-scale mechanisms that modify day-to-day variability of central-Namib FLC occurrence during different seasons.

*Code and data availability.* The ERA5 meteorological reanalysis data are freely available at the Copernicus Climate Change Service (C3S) Climate Date Store: https://cds.climate.copernicus.eu/#!/search?text=ERA5&type=dataset (last access: September 6th, 2019). Satellite data and code for data processing are available from the corresponding author upon reasonable request.

## Appendix A: Free tropospheric moisture transport and temperature anomalies

5 ## Appendix B: Equations of statistical validation measures

$$\text{POD} = \frac{a}{a+c}$$
$$\text{PC} = \frac{a+d}{a+b+c+d}$$
$$\text{FAR} = \frac{b}{a+b}$$
$$\text{CSI} = \frac{a}{a+b+c}$$
10 $$\text{BS} = \frac{a+b}{a+c}$$
$$\text{HSS} = \frac{2(ad-bc)}{(a+c)(c+d)+(a+b)(b+d)}$$

with a = number of hits, b = number of false alarms, c = number of misses and d = number of correct negatives

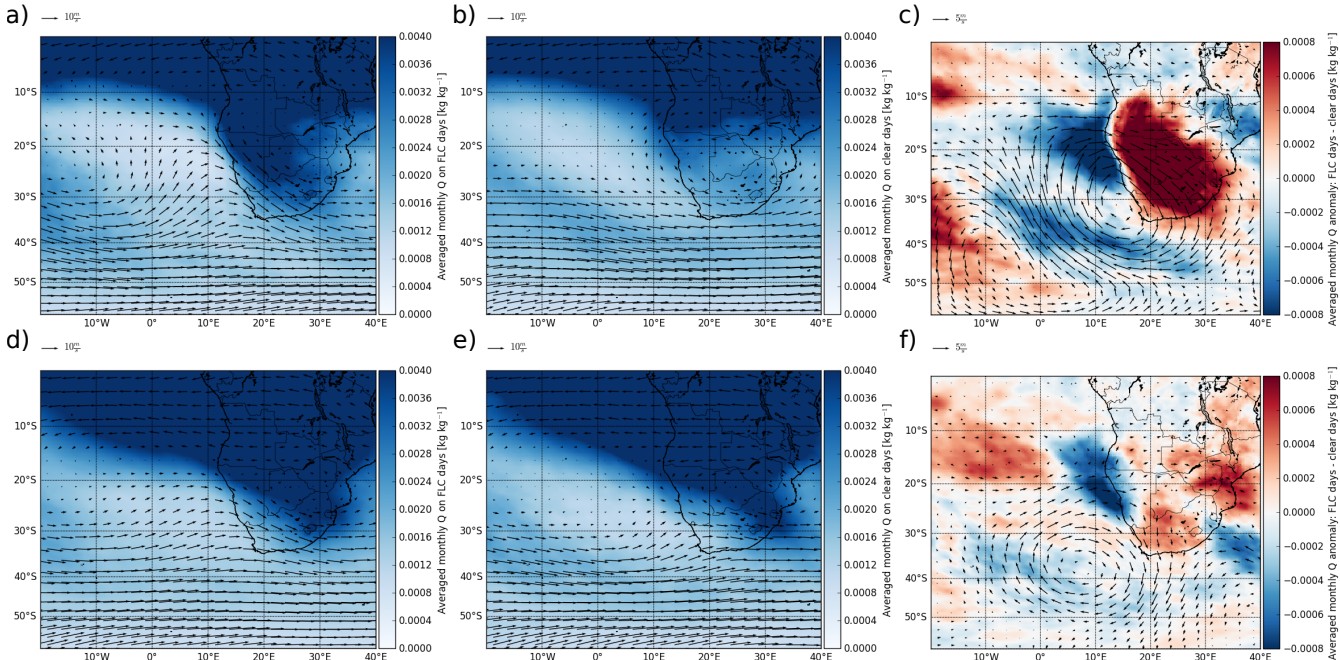

**Figure A1.** Seasonal averages of Q and winds at 700hPa on FLC days (left), clear days (center), and their difference (right) during AMJ (top) and SON (bottom).

*Author contributions.* HA and JC had the idea for the analysis. HA obtained and analyzed most of the data sets, conducted the original research and wrote the manuscript. JC and JF contributed to the study design, and JF computed initial backtrajectories. PK helped to develop a conceptual understanding of the synoptic-scale patterns and physical mechanisms. JQ computed the backtrajectories with Lagranto. MG conducted the PCA analysis. SS contributed to the design of the statistical model, and RV contributed insights to local-scale processes. JC, JF, PK, JQ, MG, SS and RV contributed to manuscript preparation, and the interpretation of findings.

*Competing interests.* The authors declare that they have no conflict of interest.

*Acknowledgements.* Funding for this study was provided by Deutsche Forschungsgemeinschaft (DFG) in the project Namib Fog Life Cycle Analysis (NaFoLiCA), CE 163/7-1. HA acknowledges receiving a Research Travel Grant by the Karlsruhe House of Young Scientists that supported a stay at ETH Zürich and thus facilitated the collaboration with SS. The contribution of JQ was funded by the Helmholtz Association (grant VH-NG-1243) MG was supported by the French National Research Agency under grant agreement no ANR-15-CE01-0014-01 (AEROCLO-sA). We acknowledge support by the KIT-Publication Fund of the Karlsruhe Institute of Technology. We thank the three anonymous reviewers for their valuable and constructive comments.

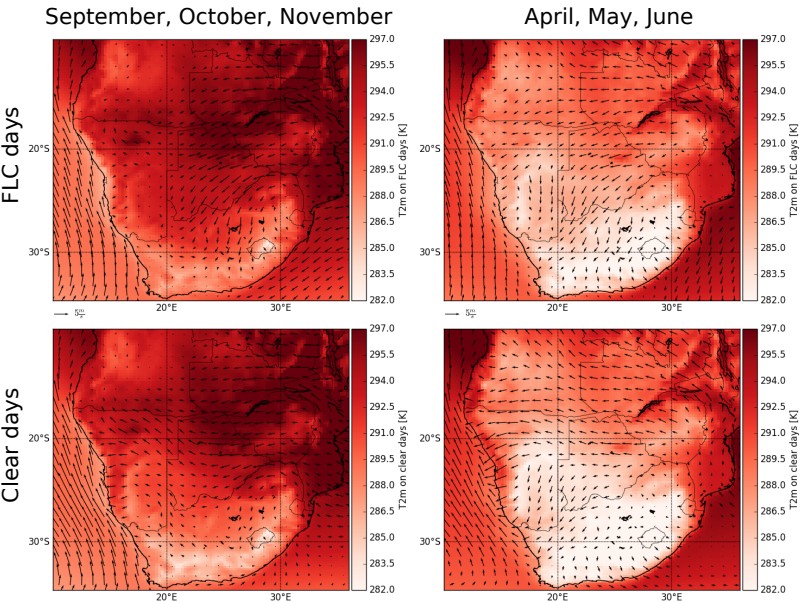

**Figure A2.** Seasonal average ERA5 T2m and 10m winds in SON (left-hand panels) and AMJ (right-hand panels) for FLC (top) and clear (bottom) days. Winds are averaged to a 1°x1° resolution.

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
