# Peer review of "Synoptic-scale controls of fog and low cloud variability in the Namib Desert"

_Atmospheric Chemistry and Physics, 2019_

## Referee Comment (RC1) · Anonymous Referee #1 · 19 Nov 2019

Review of "Synoptic-scale controls of fog and low clouds in the Namib Desert" by Andersen et al.

Using a 14-year period of reanalysis grids and backward trajectories, this study examines the impact of large-scale dynamics and thermodynamics on fog and low clouds (FLCs) over Namib. Specifically, the authors' focus on two seasons when different FLC types are observed due to different synoptic-scale regimes. A main finding is that the mean sea level pressure (MSLP) field differs notably between clear and FLC days. To this end, the authors' use a statistical model and MSLP fields to provide skillful prediction of FLCs up to one day in advance. A new conceptual model of the two different FLC regimes is developed to summarize findings and aid in future studies related to FLCs over Namib. In general, the scientific purpose is justified, the findings are im-

portant, and the paper is well-written; however, I do have concerns about some of the methods used. Overall, I think that the results are interesting and worthy of publication, and at this stage I suggest acceptance subject to major revisions.

Major/general comments: 1. Use of MSLP, 2 m temperature, and 10 m winds to characterize synoptic-scale conditions This study relies on the assumption that near-surface (boundary layer) meteorological variables – specifically MSLP, 2 m temperature, and 10 m horizontal wind components – are representative of the large-scale dynamics. While this assumption may be justified over the ocean, it is likely not justified over land, and especially where topography is pronounced. The authors' do acknowledge this sentiment (P7, L11-13); however, I think that this consideration is more important than they suggest. In fact, the authors' even cite two different papers on P7, L6-7 that suggest that "In the Namib Desert, thermally and topographically induced local wind systems within the boundary layer modulate these synoptic air-flow patterns, and the significance of the induced diurnal oscillations can exceed that of the synoptic scale". To this end, the authors' should also examine the aforementioned dynamic and thermodynamic variables at other (isobaric) levels (e.g., 925 hPa and 850 hPa) because i) the assumption of a standard atmosphere will be required for fewer locations (compared to estimating MSLP) and ii) the influence of local terrain will be suppressed at more locations. While the main conclusions of this study should not change notably, it will be interesting to see how much the PCA and statistical model results differ when using e.g., 925 hPa or 850 hPa fields. These results should be of interest to both the research and operational forecasting communities. Moreover, the impact of using the isobaric fields should be included in the context of Sections 3.1, 3.2, and 3.3: whether considering these isobaric fields is important when relating synoptic-scale meteorology to FLC occurrence.

Minor/specific comments: 1. P1, L7: When you say "significantly", do you mean in the statistical sense? If so, please specify this. If not, please choose different wording.

2. P3, L14: Please provide the retrieval wavelength(s) of the SEVIRI data used in this

study.

3. P3, L20-21: Why use these criteria? Are they following a previous study?

4. P4, L9: Which "different pressure levels" are used?

5. P5, L3-4: What is the justification for using 0.5 deg rather than 0.25 deg ERA5 grids?

6. P5, L13-15: Please provide references for the PCA method.

7. P5, L19: What is the reasoning for remapping the wind fields to a 2 deg grid?

8. P5, L19-20: Please explain why the temporal – rather than the spatial – anomalies are used. Was care taken to ensure that this 14-year period is not anomalous in some way? A 14-year sample is likely not long enough to capture some of the climatological signals at a given location. I would think that spatial anomalies would be more appropriate.

9. P5, L25-26: Please make it explicitly clear that the statistical model in this study will use spatial patterns of pressure fields.

10. P6, L19-20: What is the percentage of data availability?

11. P6, L22: I do not understand why 0.25 deg grids are used for the statistical model and coarser grids are used for other portions of the analysis. Please explain.

12. P6, L29: For readers who may be unfamiliar with the St. Helena High and the southern African continental high, please provide references. Also, is the St. Helena High over the ocean? Please add some detail here.

13. P7, L9: Do you mean thermal stability?

14. P8, L4: To which trough are you referring? This is the first time that a trough is mentioned.

15. P8, L4: "Z500 on FLC days" – please refer to the panel to help the reader.

16. P9, L2: Do you mean significant at the 0.01 level?

17. P9, L8-9: I am not sure that I understand this explanation of the dry slot. Is it possible that TCWV is reduced simply because at low levels water vapor is condensed into liquid water as FLCs form? Examining vertical profiles of TCWV may help clarify.

18. P9, L10-11: The strongest positive 2 m temperature anomalies are shifted west of the strongest positive TCWV anomalies. Can you explain why this pattern is observed?

19. P9, L11-12 & L21; P10, L2-3: These statements about greenhouse warming are a bit speculative and should be fleshed out with additional discussion/analysis/evidence. Is it possible to look at vertical profiles of heat fluxes/heating rates?

20. P9, L28: Please provide a citation for this statement.

21. P10, Fig. 3: I recommend making the contours of significant differences a different color because at present they are difficult to discern from the country boundaries.

22. P10, L3-4: Analysis of vertical profiles may help clarify and substantiate this claim.

23. P11, L7-11: Please reference Fig. 3 here.

24. P14, Fig. 6 panel b: Are you able to say something about the offshore Q anomalies in AMJ? Why do we see the positive Q anomalies increase in height farther away from the shoreline?

25. P15, Fig. 7: Can you estimate the absolute value of the pressure where the backtrajectories are initialized (25 hPa above ground level)? This will help the reader understand how much the parcels are traversing in the vertical.

26. P16, L3: The material in this paragraph does not seem to fit with the other material in this section. Perhaps improve the connection, create a new section and flesh out, or add to a different section.

27. P20, L3: Relative humidity or specific humidity?

Grammatical/wording recommendations: 1. P14, L15: Please change "along all those backtrajectories" to "along all of the backtrajectories".

---

## Referee Comment (RC2) · Anonymous Referee #2 · 21 Nov 2019

Review of "Synoptic-scale controls of fog and low clouds in the Namib Desert" by
Hendrik Andersen, et al., submitted to ACPD

In this study of fog and low-cloud (FLC) frequency in the central Namib coastal desert, the authors first present a novel 14 year satellite climatology (originally published in Andersen et al., 2019) of a relatively small region (~20,000 km$^2$). Then they select the most and least foggy days (amounting to about half of the total observations) in the two transition seasons (Fall and Spring), neither of which is the FLC frequency maximum, and then present the synoptic conditions based on reanalysis data under which foggy vs. clear days present.

The writing is clear and the figures are exceptional, however I find the inferences of causation to be quite speculative and not very convincing. I appreciate the observational nature of the analysis, and would not suggest that modeling needs to accompany it.  However, the assertions, such as radiative cooling in the more arid lower troposphere somehow being the driving factor in determining fog presence, needs to have some quantitative basis – or at the very least make reference to some other studies that have shown this effect to be important. I would be surprised if a change of a few kg m$^{-2}$ of water vapor was able to lead to increased radiative cooling rates of greater than ~0.5 K/day at the very most.  Is this sufficient to dominate the influences that create foggy conditions?  I am not sure, but without any reference to other work that may have found this to be true, it holds the scientific merit of nothing more than pure speculation. Therefore, I have a hard time seeing that this work can in the words of the authors bring about "a new conceptual model of the synoptic-scale mechanisms that control fog."

One of the stark shortcomings of this work is the absence of a lot of FLC work that has been done in other eastern basin upwelling systems, which could shed a lot of light on the interpretation and analysis of this work.  For example, the relationship between fog (or marine stratocumulus) and subsidence is completely overlooked, despite there being ample correlations pointed out in the literature (see, for example, Bony & Dufresne, 2005). Meanwhile lower tropospheric stability (LTS) is presented in Figures 3 & 5, but not really discussed at all. Other conspicuously missing prerequisite work includes Clemesha et al. (2017), Iacobellis & Cayan (2013), Koračin et al. (2005), and Dorman et al. (2019) to name a few. Furthermore, not nearly enough emphasis is paid to the effects of upwelling on the SST's and the SST anomalies on the fog. This is especially surprising given that a large portion of what controls upwelling is coastal geography which influences the wind curl along the coast (see Koračin et al., 2004).

I do not wish to sound too damning in my criticism of the work being pure speculation, but let me propose an entirely different interpretation of the data in this paper that would construct a competing narrative, or conceptual model, of the synoptic controls on coastal fog. To wit, enhanced negative vorticity advection upwind of the target site on foggy days induces subsidence which increases LTS, drying the lower troposphere, reducing marine boundary layer (MBL) entrainment, increasing surface winds and thus latent heat fluxes from the ocean, and allowing for greater moisture build-up in the MBL prior to encountering the lowest SST's of the upwelling system along the coast.

In light of the speculative nature of the manuscript as it stands, and that the value of the climatology has already been made available to the community (in Andersen et al., 2019), I would recommend not publishing this without major revisions in order to substantiate the conceptual model of fog production presented herein.

Specific Comments are presented below in order of appearance:

p.1, l.6:  It is not clear why these two seasons are chosen.  AMJ is not a common seasonal breakdown either – it is late fall into winter.  What is meant by "characterize seasonal fog" exactly?

Figure 1:  First a clarification - 1c) shows the average FLC occurrence over all days (from 3-9 UTC), and the peak is during the SH summertime, is that correct?

Also, I wonder about the wisdom of fixing this time window rigidly past the falling edge of the fog 'burn off'.  Sunrise times in that area shift from ~4:00 UTC in summer to 5:45 UTC in winter, which is an appreciable portion of this 6 hr window.  I worry that this could bias the FLC frequency changes observed by season.

I think it might be useful to compare your results to any other cloud climatologies that exist for the region. For example, Dorman et al., 2019 present a COADS-based fog climatology that suggests a fog peak in MAM months in the Benguela upwelling system.

I think the monthly FLC pattern is central enough to this work to warrant a line graph as opposed to this subtle gray scale figure which allows for a much less quantitative comparison of the seasons.

Finally, it seems to me if you are going to carry out an annual analysis of FLC-Clear (as you do in Fig 3), you need to report what fraction of your clear and FLC days from your histogram come from each season.  Because the pattern you see in Fig. 3 could match the patterns you see in Figs. 4/5 for SON simply because that is where the majority of your FLC days throughout the year come from.

p.7, l.15:  This is confusing because you are focusing on SON, and only the thin latitudinal band from ~22-24°S, the FLC peak actually occurs in DJF (as shown in Fig. 1c & Andersen 2019, fig. 2c.)

p.8, l.3:  The winds are southerly throughout the region, how do you infer "northerly" advection?

p.8, l.7:  You are referring to features of the climatological Z500 pattern without showing what that is, so it is hard to assess these statements about a trough and the absence of a coastal low.

Are you sure Olivier and Stockton (1989) are not referring to a particular time of year for their coastal trough as opposed to a year round analysis that you are presenting here?
A quick look at NCEP reanalysis data for the region shows a subtle trough upwind of the coastline.

[Figure]

p.9, l.4:  I think this SST time lag inference is unfounded speculation on the authors' part.  The wind difference indicates to me that the clear days have slightly stronger offshore wind components, which could weaken ocean upwelling.  It is the alongshore wind component that determines the upwelling, and could possibly have subtle variations due to coastline geography (see, for example, Koračin, Darko, Clive E. Dorman, and Edward P. Dever. "Coastal perturbations of marine-layer winds, wind stress, and wind stress curl along California and Baja California in June 1999." Journal of Physical Oceanography 34, no. 5 (2004): 1152-1173.)

p.9, l.5:  The SST anomaly having a hydrostatic impact on MSLP seems highly unlikely given that the FLC effects are associated with strong synoptic forcing as argued in the last paragraph. Furthermore, how exactly does it appear likely that SST-FLC correlations are most 'pronounced on seasonal scales'? Can't that be determined for your data set and put to the test? There is not all that much variability in SST in this region, as far as I can see from NCEP reanalysis data.

p.9, l.10:  This speculation would benefit from some sort of simple calculation of the magnitude of this effect.  Are you meaning to say that radiative cooling will be significantly influencing the SST's? If so, this seems unlikely in a strong upwelling system such as this.  Or are you saying that

FLC, once formed, will be sustained by effective cloud-top radiative cooling due to the dry tongue over it? As it stands this just seems like a qualitative speculation that is unsubstantiated (without at least a reference to another work that has explored a comparable situation, or a back of the 'envelope' calculation on your part.) The same holds for the assertion that moisture advection influences the surface heat low by principally radiative means presented in the following sentences.

p.9, l.14/15: This hypothesis could be tested by looking at the T anomaly only during the overnight hours to see if it is an air mass difference or an insolation difference (I strongly suspect it is the latter.) My hunch is that it will be slightly warmer overnight because of radiative heating of the surface from the FLC, which would provide evidence against the air mass difference hypothesis.

p.9, l.27: In the discussion surrounding the similar annual pattern of Fig. 3b you referred to it as a trough instead of a cut-off low.

p.11, l.4: A few 0.1's K is a subtle change, but the increased wind speeds could definitely increase the latent heat fluxes in the upwind region. Here, you could get a sense of the relative magnitude of these effects by using a simple moisture exchange coefficient and quantifying differences in saturation vapor pressures vs. mean wind speeds.

p.11, l.6: I would bet that it has everything to do with upwelling induced by the wind field.

p.11, l.30: Or the dry anomalies could be associated with subsidence which augments the LTS in the fog cases. This reduces MBL entrainment and along with increased LH fluxes upwind helps to build up Q in the MBL. Along with a lower SST, these influences act in tandem to reduce the dew point depression.

Figure 5: Very little attention is paid to the LTS anomalies presented. My read of Figs. 4/5 is that regardless of season FLC is strongly associated with low SSTs, low T2m, and high LTS.

 p.14, l.17: You could look at potential temperature to see what sort of effects that radiative cooling has on the foggy days. It seems that potential temperature would be a better variable to present in the back trajectories (unless, of course, it is a purely isentropic back trajectory.)

p.14, l.21 to p.15, l.1: Doesn't this contradict your hypothesis presented earlier about the lower column water vapor leading to greater radiative cooling on the foggy days?

p.18, l.19: It is not too surprising that so much is explained by the MSLP fields because they determine a lot of things. For instance, MSLP is the main variable used in calculating conventional upwelling indices. Again, I found the lack of centrality of coastal SSTs to be surprising in this work given how important it is found to be in most other studies.

---

## Referee Comment (RC3) · Anonymous Referee #3 · 5 Dec 2019

This is a well-written article. The authors are to be congratulated.

However, It has long been known that fog and low cloud in the coastal zone of the Namib and the South African west coast are largely due to a local/meso-scale phenomenon called a coastal low. This is a weak low pressure system trapped between the western escarpment to the east and the Benguela current to the east. It only extends to just above the height of the escarpment. The diameter of the coastal low and the extent of the cold water upwelling region often determines whether fog occurs or not. An interplay between an approaching cold front and a HIGH pressure system over the continent is thought to cause the coastal low and associated fog to move southwards from the Namibian coast, down the South African west coast, around the tip of South Africa and northwards towards Kwazulu-Natal. It is unclear why the authors

need to work at synoptic scale when the phenomenon occurs at a much smaller scale. The role of a cut-off low in fog occurrence is really surprising.

It is suggested that much more information is provided on the research that has already been conducted on the occurrence of fog along the southern African west coast.

---

## Author Comment (AC1) · 7 Jan 2020

**Synoptic-scale controls of fog and low clouds in the Namib Desert: Response to Reviewer 1**

**Hendrik Andersen, Jan Cermak, Julia Fuchs, Peter Knippertz, Marco Gaetani, Julian Quinting, Sebastian Sippel, and Roland Vogt**

contact: **hendrik.andersen@kit.edu**

*We would like to thank reviewer 1 for her/his careful review of the manuscript and her/his constructive criticism and valuable comments. Comments by the referee are colored in black, our replies or comments are colored in blue and italics.*

Using a 14-year period of reanalysis grids and backward trajectories, this study examines the impact of large-scale dynamics and thermodynamics on fog and low clouds (FLCs) over Namib. Specifically, the authors' focus on two seasons when different FLC types are observed due to different synoptic-scale regimes. A main finding is that the mean sea level pressure (MSLP) field differs notably between clear and FLC days. To this end, the authors' use a statistical model and MSLP fields to provide skillful prediction of FLCs up to one day in advance. A new conceptual model of the two different FLC regimes is developed to summarize findings and aid in future studies related to FLCs over Namib. In general, the scientific purpose is justified, the findings are important, and the paper is well-written; however, I do have concerns about some of the methods used. Overall, I think that the results are interesting and worthy of publication, and at this stage I suggest acceptance subject to major revisions.

Major/general comments:
1. Use of MSLP, 2 m temperature, and 10 m winds to characterize synoptic-scale conditions
   This study relies on the assumption that near-surface (boundary layer) meteorological variables – specifically MSLP, 2 m temperature, and 10 m horizontal wind components – are representative of the large-scale dynamics. While this assumption may be justified over the ocean, it is likely not justified over land, and especially where topography is pronounced. The authors' do acknowledge this sentiment (P7, L11-13); however, I think that this consideration is more important than they suggest. In fact, the authors' even cite two different papers on P7, L6-7 that suggest that "In the Namib Desert, thermally and topographically induced local wind systems within the boundary layer modulate these synoptic air-flow patterns, and the significance of the induced diurnal oscillations can exceed that of the synoptic scale". To this end, the authors' should also examine the aforementioned dynamic and thermodynamic variables at other (isobaric) levels (e.g., 925 hPa and 850 hPa) because i) the assumption of a standard atmosphere will be required for fewer locations (compared to estimating MSLP) and ii) the influence of local terrain will be suppressed at more locations. While the main conclusions of this study should not change notably, it will be interesting to see how much the PCA and statistical model results differ when using e.g., 925 hPa or 850 hPa fields. These results should be of interest to both the research and operational forecasting communities. Moreover, the impact of using the isobaric fields should be included in the context of Sections 3.1, 3.2, and 3.3: whether considering these isobaric fields is important when relating synoptic-scale meteorology to FLC occurrence.

*Thank you for this comment. In the manuscript, we use Z500 as the classic weather characteristic that is thought to represent the synoptic circulation in the free troposphere. MSLP and 10 m winds are used to describe the topography-near circulation. For MSLP and 10 m winds, caveats exist, as pointed out by the reviewer and also briefly discussed in the orignial manuscript on P7 L 11-13. Due to the high topography in southern Africa, and the fact that the region of interest (central Namib) lies close to the coast, and at low altitudes, no one specific pressure level can adequately summarize near-surface conditions throughout this large domain. In the preparation of the original manuscript, we carefully investigated more pressure levels than presented (1000 to 500 hPa in 100 hPa steps). In Fig. R1.1 of this response, we compare the anomaly patterns in Z850 (top), Z925 (middle) with those of MSLP (bottom), following your suggestion. It is clearly apparent that the relevant anomaly patterns are not affected in a significant way by the choice of the pressure level within the lower troposphere. The biggest difference between the investigated levels are the wind anomalies north of the central Namib (star), where 10 m wind anomalies are small, but at higher levels on FLC days, a marked onshore anomaly exists. This may be an indication of a blocking of onshore flow below the inversion, and a more freely flowing air at pressure levels above 925 hPa.*

[Figure]

*Fig. R1.1: Averaged monthly mean differences (FLC days - clear days) in Z850 and 850 hPa winds (top), 925 hPa (middle), and MSLP (bottom) with 10m winds for SON (left-hand panels) and AMJ (right-hand panels). Contours mark regions where the distributions differ significantly at the 0.01*

*level. u and v vectors of winds are interpolated bilinearly interpolated to a 2.5° x 2.5° grid for clarity.*

*In section 3.1, we extended our discussion on this:*
*"However, due to the joint consideration of regions in southern Africa with high topography, the low-lying central Namib, and marine regions, no one specific pressure level of geopotential height can adequately summarize near-surface conditions throughout this large domain. Additional analyses show that difference patterns obtained from MSLP fields in southern Africa are similar to those at 925 hPa and 850 hPa (not shown)."*
*In section 3.2, we additionally discuss this in the updated version of the manuscript:*
*"As discussed in Sec. 3.1, MSLP and 10m winds may not be a good representation of near-surface level characteristics where topography is high, however, additional analyses of geopotential height at 850 hPa and 925 hPa corroborate the observed MSLP patterns. Differences exist in winds north of the central Namib, where at 925 hPa and 850 hPa (not shown), a stronger onshore flow anomaly is observed than at 10 m, possibly indicating a topographical blocking of the onshore flow below the inversion."*

*Based on your suggestion, we also ran the ridge regression model to predict FLC days and clear days based on Z850 and Z925. We find that the skills of the different models based on MSLP, Z850, and Z925 are very similar, with MSLP a slightly better predictor (Tab. 1 of this document). In section 3.5 of the updated version of the manuscript, we now state:*
*"As MSLP fields in southern Africa may not be representative due to the high topography, the model was additionally run based on Z850 and Z925 fields. The model performances were nearly the same (overall PC of 84 % in both cases), suggesting that it is adequate to use MSLP in this context."*

*Tab. 1: The performance of the ridge model to predict FLC and clear days based on three different sets of predictors (MSLP, Z850, Z925).*

|  | POD | FAR | PC | BS |
|---|---|---|---|---|
| **MSLP** | 0.94 | 0.17 | 0.86 | 1.14 |
| **Z850** | 0.90 | 0.18 | 0.84 | 1.10 |
| **Z925** | 0.91 | 0.19 | 0.84 | 1.12 |

Minor/specific comments:

1. P1, L7: When you say "significantly", do you mean in the statistical sense? If so, please specify this. If not, please choose different wording.
   *The sentence now says:*
   *" It is found that during both seasons, mean sea level pressure and geopotential height at 500 hPa differ markedly between fog/low-cloud and clear days, [...]".*

2. P3, L14: Please provide the retrieval wavelength(s) of the SEVIRI data used in this study.
   *This is now included in the updated version of the manuscript. In section 2.1, we now state that the*
   *"algorithm relies mostly on a channel difference in the thermal infrared (12.0-8.7µm), [...]".*

3.  P3, L20-21: Why use these criteria? Are they following a previous study?
    *This is now described in more detail in section 2.1. The updated version of the manuscript states:*
    *"A specified averaging time period is needed to avoid statistically mixing two separate FLC events occurring on successive nights which would be the case in a daily average FLC occurrence data set. The specific time period is chosen to include all periods of the diurnal cycle, with FLC occurrence rising, peaking, and starting to dissipate (Andersen and Cermak, 2018) during this time."*

4.  P4, L9: Which "different pressure levels" are used?
    *We now provide the information on all pressure levels used. In section 2.2 of the updated version of the manuscript, we now state that:*
    *"To characterize large-scale dynamic and thermodynamic conditions, fields of mean sea level pressure (MSLP), geopotential height at 500, 700, 850, and 925 hPa (Z500, Z700, Z850, Z925), 2 m air temperature (T2m), sea surface temperature (SST), total columnar water vapor (TCWV), specific humidity (Q), as well winds at 10m and at all ERA5 pressure levels between 1000 and 500 hPa, and lower tropospheric stability (LTS: computed as the difference between potential temperature at 700 hPa and T2m (Klein and Hartmann, 1993)) are used. To represent the morning conditions for which FLC is averaged, 6 UTC fields of ERA5 data are selected. While for additional analysis, T2m fields are also used at nighttime (1 UTC and 3 UTC), the 6 UTC fields are used if no specific information on time is given."*

5.  P5, L3-4: What is the justification for using 0.5 deg rather than 0.25 deg ERA5 grids?
    *The computational cost and data storage (with thoe chosen resolution, the ERA5 data already require ~30TB of disk storage). Clearly, 0.25 degrees would be preferable, but we calculated many more backtrajectories than shown in the manuscript (different seasons, times, initial altitudes), and so this really became an issue. We do not expect the clear differences between FLC and clear days to be substantially affected, though. We initially calculated the backtrajectories with the HYSPLIT model based on ERA-Interim data, which are shown in Fig. R1.2 of this response (compare with Fig. 7 of the original manscript). We now state in section 2.3:*
    *"The spatial resolution is used to reduce the data volume and computational cost. While the native resolution would be preferable, the general patterns of the trajectories are not expected to be affected, as tests with lower-resolution ERA-Interim data showed comparable results."*

[Figure]

*Fig. R1.2: Hysplit backtrajectories for SON (left-hand panels) and AMJ (right-hand panels) for FLC days (top) and clear days (bottom), based on ERA-Interim data at 0.75° spatial resolution.*

6.  P5, L13-15: Please provide references for the PCA method.
    *The reference for the PCA method is Storch and Zwiers (1999), page 5, line 17 in the first submitted version. We moved the reference to the beginning of Sec. 2.4 for clarity.*

7.  P5, L19: What is the reasoning for remapping the wind fields to a 2 deg grid?
    *We first remapped data to 2 degrees for computational reasons, because PCAs are quite computationally expensive (Pham Thanh et al. 2019), and the Matlab software used to compute the PCA is unable to do this for a 14-year daily time series (365x14=5110 time steps) at such high resolution (40x40 degree = 160x160=25600 grid points). Moreover, the purpose of the analysis is to detect the main modes of variability of surface wind in the region, to assess possible drivers of fog occurrence at synoptic scale. In this respect, 2 degree resolution is a reasonable compromise for capturing synoptic-scale variability and smooth out the variability associated with fine-scale effects. However, we additionally performed a new PCA analysis by regridding to 1 degree resolution, to continue to smooth out fine scale disturbances not to lose too much of the variability of the original dataset. The results of the PCA at the two resolutions are very similar. These aspects are now described in Sec. 2.4:*
    *"Remapping to 1° resolution allows to accurately describe the atmospheric variability at synoptic scale, but smoothing out the variability associated with small-scale effects. The sensitivity of the PCA to the spatial resolution is tested by conducting the analysis based on wind fields remapped to a 2° resolution. The results of the two PCAs at the different resolutions are very similar, demonstrating their robustness. Daily anomalies are computed with respect to the 14-year sampling of the FLC dataset in order to compare wind and FLC variability over a homogeneous climatology."*

8.  P5, L19-20: Please explain why the temporal – rather than the spatial – anomalies are used. Was care taken to ensure that this 14-year period is not anomalous in some way? A 14-year sample is likely not long enough to capture some of the climatological signals at a given location. I would think that spatial anomalies would be more appropriate.
    *Thanks for the comment, which gives us the opportunity to further clarify the PCA approach used. The aim of the analysis is first to identify the main modes of variability of the near-surface circulation, then to relate the selected modes to FLC occurrence, to assess possible correlations, i.e. to assess how the circulation patterns explain FLC variability. In practice, we need to correlate daily time series associated with wind EOFs and FLC occurrence. Therefore, by removing the climatological mean we require that the covariance matrix is equal to the temporal variance matrix. The computation of such temporal anomalies is a common preprocessing step in PCA analysis (see e.g. Gaetani et al. (2016) or Pham Thanh et al. (2019) for other examples). We clarified this aspect in the revised version in Sec. 2.4 and now discuss this in the conclusions:*
    *"While a 14-year sample is not optimal to capture climatological variability, the mechanisms documented here for the first time are unlikely to be fundamentally different in other climatological periods."*

9.  P5, L25-26: Please make it explicitly clear that the statistical model in this study will use spatial patterns of pressure fields.
    *In these lines of the text, the statistical model used (ridge regression) is not described. The description of how the ridge regression is applied is found in P6 L19-22. Here, we now explicitly state that spatial patterns are used:*
    *"[...] using spatial patterns of 6 UTC (representative of averaging time of FLC cover, see Sec. 2.1) ERA5 MSLP fields[...]"*

10. P6, L19-20: What is the percentage of data availability?
    *Maybe this is a misunderstanding: The statement "for which observations exist" refers to the period in time that the satellites were/are in orbit. This is now clarified in the updated version of the manuscript:*
    *"The ridge regression method is used to predict FLC and clear days over the complete 14-year time series,[...]"*

11. P6, L22: I do not understand why 0.25 deg grids are used for the statistical model and coarser grids are used for other portions of the analysis. Please explain.
    *For each step of the analysis, we chose the highest resolution that is feasible for the analysis. This is now explained in the updated version of the manuscript where applicable. The ridge regression is computionally cheap in comparison to the PCA analysis, allowing us to use the 0.25 resolution. Also, data storage is not an issue, because the statistical model uses just one parameter as input (e.g., MSLP). The backtrajectories, for example, rely on three wind components, as well as the temperature and humidity fields at 137 levels. We now discuss this briefly in each respective section.*

12. P6, L29: For readers who may be unfamiliar with the St. Helena High and the southern African continental high, please provide references. Also, is the St. Helena High over the ocean? Please add some detail here.
    *We have adjusted the terminology describing the pressure system as South Atlantic High and continental high for clarification.*

13. P7, L9: Do you mean thermal stability?
    *Yes, as described e.g. by the lower tropospheric stability shown in Fig. 2 of the original version of the manuscript. This is clarified in the updated version of the manuscript: "The combination of large-scale subsidence and low SSTs along the coastline produces high LTS conditions, [...]"*

14. P8, L4: To which trough are you referring? This is the first time that a trough is mentioned.
    *The absolute fields at 700 hPa are now included in the appendix of the updated manuscript, showing the trough.*

15. P8, L4: "Z500 on FLC days" – please refer to the panel to help the reader.
    *Done in the updated version of the manuscript.*

16. P9, L2: Do you mean significant at the 0.01 level?
    *No, the 0.01 level is shown in the figure, but we also computed the 0.05 level. We clarify this in the updated version of the manuscript:*
    *"There is a coherent pattern of slightly lower SSTs (~0.5 K; Fig. 3 e)) along the coastline on FLC days; however, the difference between SSTs on FLC and clear days is not significant at the 0.01 level (and also not at the 0.05 level)."*

17. P9, L8-9: I am not sure that I understand this explanation of the dry slot. Is it possible that TCWV is reduced simply because at low levels water vapor is condensed into liquid water as FLCs form? Examining vertical profiles of TCWV may help clarify.
    *We investigate the vertical moisture anomalies in Fig. 6 of the manuscript. The dry anomaly is fairly deep, from the top of the MBL to 600 hPa in AMJ, and extending higher than 500 hPa in the SON. The region is very stable, and the MBL moisture is expected to be driven from surface fluxes. Fig. R1.3 in this response shows seasonally averaged Q and winds at 700 hPa, the layer with the strongest Q difference, for FLC days (left), clear days (center), and their difference (right). During both, AMJ (top) and SON (bottom), it is clearly visible how the synoptic-scale disturbance induces a horizontal transport of drier air over the study region. During AMJ, the free-tropospheric moisture transport into continental southern Africa is clearly visible. We now include this figure in the appendix of the manuscript.*

[Figure]

*Fig. R1.3: Seasonal averages of Q and winds at 700hPa on FLC days (left), clear days (center), and their difference (right) during AMJ (top) and SON (bottom).*

18. P9, L10-11: The strongest positive 2m temperature anomalies are shifted west of the strongest positive TCWV anomalies. Can you explain why this pattern is observed?
*The temperature anomalies are also caused by warm air advection near the surface, but during AMJ, when the moist anomaly is pronounced, the spatial patterns of TCWV and T2m agree quite well. This is shown by Fig. R1.4 of this document, which shows the relationship of T2m anomalies and TCWV anomalies in continental regions where T2m is significantly higher on FLC days than on clear days. A clear relationship is obvious, with a correlation coefficient of 0.75. This statistical relationship indicates that more than half of the observed T2m anomalies can be explained by the TCWV anomalies, underscoring the relevance of the greenhouse effect for T2m during this time. We now state in section 3.3 that:*
*"Here, the T2m anomalies closely follow those of TCWV (Pearson correlation coefficient of 0.75 in continental regions with significantly higher T2m on FLC days than clear days), suggesting that the increased moisture causes an additional surface heating due to greenhouse warming as discussed in Sec. 3.2."*

[Figure]

*Fig. R1.4: The relationship of TCWV anomalies and T2m anomalies over land during AMJ, and in regions where T2m is statistically significantly higher on FLC days than on clear days.*

*Fig. R1.5 of this document shows average T2m and 10m winds during FLC and clear days during the different seasons, illustrating the warm-air advection on FLC days that also contributes to the overall anomaly pattern. The figure is now in the appendix of the updated version of the manuscript.*

[Figure]

*Fig. R1.5: Seasonal average ERA5 T2m and 10m winds in SON (left-hand panels) and AMJ (right-hand panels) for FLC (top) and clear (bottom) days. Winds are averaged to a 1°x1° resolution, the considered time period is 2004-2017.*

19. P9, L11-12 & L21; P10, L2-3: These statements about greenhouse warming are a bit speculative and should be fleshed out with additional discussion/ analysis/ evidence. Is it possible to look at vertical profiles of heat fluxes/heating rates?
*We agree that we do not give quantitative estimations for heating rates, however, this effect has been shown and quantified before in the Kalahari (Manatsa and Reason (2017)), and in other dry subtropical deserts (e.g. the Sahara: (Evan et al. 2015, Alamirew et al. 2018), or the Sahel (Oueslati et al. 2017)). Also, the strong relationship of the spatial anomaly patterns of TCWV and T2m (shown in Fig. R1.4 of this document) during AMJ underscores the importance of TCWV for T2m. We now discuss this in more detail in the current version of the manuscript (Sec. 3.2):*
*"This effect of free-tropospheric moisture on surface temperatures has been observed in the Kalahari (Manatsa and Reason, 2017) and other arid and semi-arid regions before (Evan et al., 2015; Oueslati et al., 2017; Alamirew et al., 2018)."*

20. P9, L28: Please provide a citation for this statement.
*After reviewing the absolute fields of Z500 and Z700, we conclude that while the atmospheric wave is quite steep during AMJ, it does not 'break'. As such, we have deleted the notation of rossby waves and cut-off lows from the manuscript, and refer to them as synoptic-scale disturbances of different magnitude.*

21. P10, Fig. 3: I recommend making the contours of significant differences a different color because at present they are difficult to discern from the country boundaries.
*In the maps the significance is now indicated by a thicker grey line to be more clearly distinguishable from country boundaries.*

22. P10, L3-4: Analysis of vertical profiles may help clarify and substantiate this claim.
    *As outlined above, the vertical profiles are analyzed in Fig. 6 of the manuscript. In the updated version of the manuscript, we point the reader to this section earlier for clarity. This is also supported by Fig. R1.3 of this response, which is now in the appendix of the updated version of the manuscript.*

23. P11, L7-11: Please reference Fig. 3 here.
    *Yes, we have included the correct Fig. reference for clarity.*

24. P14, Fig. 6 panel b: Are you able to say something about the offshore Q anomalies in AMJ? Why do we see the positive Q anomalies increase in height farther away from the shoreline?
    *This is an interesting point. We agree that this should be discussed in more detail: Fig. R1.6 of this response shows the Q and wind difference at 950 (left) and 900 (right) hPa between FLC and clear days for AMJ. The moist anomaly in the MBL is clearly a synoptic-scale feature tied to the main disturbance. It is likely related to the cold front of the disturbance. In the updated version of the manuscript, we now state that:*
    *" During both seasons, the marine boundary layer features an onshore flow anomaly and is more humid on FLC than on clear days, especially during AMJ, where this is a synoptic-scale feature, likely related to the cold front of the disturbance."*

[Figure]

*Fig. R1.6: Difference in Q and winds at 950 and 900 hPa during AMJ.*

25. P15, Fig. 7: Can you estimate the absolute value of the pressure where the backtrajectories are initialized (25 hPa above ground level)? This will help the reader understand how much the parcels are traversing in the vertical.
    *Yes, the backtrajectories are initialized just below 940 hPa (here the backtrajectories meet in Fig. 8 d)). This is now mentioned in section 2.3 of the updated version of the manscript.*

26. P16, L3: The material in this paragraph does not seem to fit with the other material in this section. Perhaps improve the connection, create a new section and flesh out, or add to a different section.
    *Yes, we agree that the linkage of this section deserved to be improved. In the updated version of the manuscript this paragraph now starts like this:*
    *"The analysis of air-mass backtrajectories shows that the discrimination between FLC and clear days is not possible using dynamics alone, and that seasonal differences exist in the link between the probability of FLC days and advection patterns. To further investigate the role of different dynamical regimes for FLC occurrence, a PCA is conducted [...]"*

27. P20, L3: Relative humidity or specific humidity?
*This refers to the specific humidity increase shown in Fig. 6, but increase in relative humidity is also expected. We now describe this in more detail in the updated version of the manuscript:*
*"A significant pattern of SST anomalies is found only in AMJ, with anomalously high SSTs off the coast possibly acting together with increased near-surface winds to enhance surface latent heat fluxes that may contribute to the observed higher levels of specific humidity in the marine boundary-layer."*

28. Grammatical/wording recommendations: 1. P14, L15: Please change "along all those backtrajectories" to "along all of the backtrajectories".
*This is corrected in the updated version of the manuscript.*

**References**

*Alamirew, N. K., Todd, M. C., Ryder, C. L., Marsham, J. H., and Wang, Y.: The early summertime Saharan heat low: Sensitivity of the radiation budget and atmospheric heating to water vapour and dust aerosol, Atmospheric Chemistry and Physics, 18, 1241–1262, https://doi.org/10.5194/acp-18-1241-2018, 2018.*

*Andersen, H. and Cermak, J.: First fully diurnal fog and low cloud satellite detection reveals life cycle in the Namib, Atmospheric Measurement Techniques, 11, 5461–5470, https://doi.org/10.5194/ amt-11-5461-2018, https://www.atmos-meas-tech-discuss.net/amt-2018-213/, 2018.*

*Evan, A. T., Flamant, C., Lavaysse, C., Kocha, C., and Saci, A.: Water vapor-forced greenhouse warming over the Sahara desert and the recent recovery from the Sahelian drought, Journal of Climate, 28, 108–123, https://doi.org/10.1175/JCLI-D-14-00039.1, 2015.*

*Klein, S. A. and Hartmann, D. L.: The Seasonal Cycle of Low Stratiform Clouds, Journal of Climate, 6, 1587–1606, https://doi.org/10.1175/1520-0442(1993)006<1587:TSCOLS>2.0.CO;2, 1993.*

*Manatsa, D. and Reason, C.: ENSO–Kalahari Desert linkages on southern Africa summer surface air temperature variability, International Journal of Climatology, 37, 1728–1745, https://doi.org/10.1002/joc.4806, 2017.*

*Oueslati, B., Pohl, B., Moron, V., Rome, S., and Janicot, S.: Characterization of heat waves in the Sahel and associated physical mechanisms, Journal of Climate, 30, 3095–3115, https://doi.org/10.1175/JCLI-D-16-0432.1, 2017.*

*Pham-Thanh, H., Linden, R., Ngo-Duc, T., Nguyen-Dang, Q., Fink, A. H., and Phan-Van, T.: Predictability of the rainy season onset date in Central Highlands of Vietnam, International Journal of Climatology, p. joc.6383, https://doi.org/10.1002/joc.6383, https://onlinelibrary.wiley.com/doi/abs/10.1002/joc.6383, 2019*

---

## Author Comment (AC2) · 7 Jan 2020

**Synoptic-scale controls of fog and low clouds in the Namib Desert: Response to Reviewer 2**

**Hendrik Andersen, Jan Cermak, Julia Fuchs, Peter Knippertz, Marco Gaetani, Julian Quinting, Sebastian Sippel, and Roland Vogt**

contact: **hendrik.andersen@kit.edu**

*We would like to thank reviewer 2 for her/his careful review of the manuscript and her/his valuable comments, thoughts and the constructive criticism. Comments by the referee are colored in black, our replies or comments are colored in blue and written in italics. We give initial responses in the introductory text, but in the applicable cases, a more detailed discussion follows in the point-by-point responses to the specific comments.*

In this study of fog and low-cloud (FLC) frequency in the central Namib coastal desert, the authors first present a novel 14 year satellite climatology (originally published in Andersen et al., 2019) of a relatively small region (~20,000 km$^2$). Then they select the most and least foggy days (amounting to about half of the total observations) in the two transition seasons (Fall and Spring), neither of which is the FLC frequency maximum, and then present the synoptic conditions based on reanalysis data under which foggy vs. clear days present.

The writing is clear and the figures are exceptional, however I find the inferences of causation to be quite speculative and not very convincing. I appreciate the observational nature of the analysis, and would not suggest that modeling needs to accompany it. However, the assertions, such as radiative cooling in the more arid lower troposphere somehow being the driving factor in determining fog presence, needs to have some quantitative basis – or at the very least make reference to some other studies that have shown this effect to be important. I would be surprised if a change of a few kg m$^{-2}$ of water vapor was able to lead to increased radiative cooling rates of greater than ~0.5 K/day at the very most. Is this sufficient to dominate the influences that create foggy conditions? I am not sure, but without any reference to other work that may have found this to be true, it holds the scientific merit of nothing more than pure speculation. Therefore, I have a hard time seeing that this work can in the words of the authors bring about "a new conceptual model of the synoptic-scale mechanisms that control fog."

*Thank you for this statement, which made clear to us that the scope of both aims and findings was open to misinterpretation and misunderstanding.*

*On radiative cooling being 'the' driving factor: By no means do we state that the radiative cooling is **the** dominant factor driving FLC formation (or even it being the driving force for e.g. the seasonality), but rather that it is a contributing factor that facilitates FLC occurrence and thereby influences the day-to-day variability that is in the focus of this manuscript. Also, the free-tropospheric dry anomaly is larger than the TCWV composites suggest, as some of the TCWV difference is masked by the increased moisture in the marine-boundary layer on FLC days so that in relative terms, the difference in specific humidity between FLC days and clear days is quite substantial (up to 220 % at ~700 hPa; P11,L29). In the discussion on the specific comment, we list and discuss studies that have shown the cooling (and also warming) effects of TCWV to be important for low-cloud cover (and also land surface temperatures) in the region.*

*On the conceptual model comment: Indeed we recognize not to have chosen optimal wording to communicate the conceptual model in the abstract: In the updated version of the manuscript, we use a more precise description (similar to that used e.g. in the caption for the conceptual model): "a new conceptual model of the synoptic-scale mechanisms that control fog and low cloud*

*variability [...]" . We also decided to change the title of the manuscript to "Synoptic-scale controls of fog and low cloud **variability** in the Namib Desert". Both changes are intended to clarify that in this study we do not analyze the drivers of the Namib-region FLC system, but the drivers of the day-to-day variability within the system.*

One of the stark shortcomings of this work is the absence of a lot of FLC work that has been done in other eastern basin upwelling systems, which could shed a lot of light on the interpretation and analysis of this work. For example, the relationship between fog (or marine stratocumulus) and subsidence is completely overlooked, despite there being ample correlations pointed out in the literature (see, for example, Bony & Dufresne, 2005). Meanwhile lower tropospheric stability (LTS) is presented in Figures 3 & 5, but not really discussed at all. Other conspicuously missing prerequisite work includes Clemesha et al. (2017), Iacobellis & Cayan (2013), Koračin et al. (2005), and Dorman et al. (2019) to name a few. Furthermore, not nearly enough emphasis is paid to the effects of upwelling on the SST's and the SST anomalies on the fog. This is especially surprising given that a large portion of what controls upwelling is coastal geography which influences the wind curl along the coast (see Koračin et al., 2004).

*Thank you for pointing us to these interesting and relevant publications, and, indeed the links to other coastal upwelling systems are now strengthened in the manuscript. We did point out on page 9, L 9-11 of the original version of the manuscript that "These stable conditions promote the formation of the southeastern Atlantic stratocumulus cloud deck and determine its seasonal cycle (Klein and Hartmann, 1993; Andersen et al., 2017)." We agree that we should discuss stability and SSTs more, as they are clearly main drivers of the FLC system. As the focus of the manuscript lies on synoptic-scale modifiers of day-to-day FLC variability within this system, and both SST and LTS difference patterns are not as marked as the other mechanisms described in this paper, we did not discuss them in similar detail. However, after carefully considering the points brought up in this review, we agree that this might actually be a shortcoming of the manuscript in its current form. Therefore we have decided to include and explain in much more detail the role of SST and LTS for FLCs in this region, and to discuss links to comparable systems, also based on the valuable sources pointed to by the reviewer. In the point-by-point responses below, we discuss this in more detail, show additional analyses and present the changes to the manuscript.*

I do not wish to sound too damning in my criticism of the work being pure speculation, but let me propose an entirely different interpretation of the data in this paper that would construct a competing narrative, or conceptual model, of the synoptic controls on coastal fog. To wit, enhanced negative vorticity advection upwind of the target site on foggy days induces subsidence which increases LTS, drying the lower troposphere, reducing marine boundary layer (MBL) entrainment, increasing surface winds and thus latent heat fluxes from the ocean, and allowing for greater moisture build-up in the MBL prior to encountering the lowest SST's of the upwelling system along the coast.

*We thank the reviewer for sharing her/his thoughts about this. You have triggered engaging and fruitful discussions among the authors. While we agree with some aspects of this alternate conceptual model, and will discuss those in more detail in the updated version of the manuscript (increased wind speeds lead to an increased upwind latent heat flux, building up moisture in the marine boundary layer, at least in AMJ; advection over the lowest SSTs), Fig. 6 in the manuscript gives no indication for a relevant difference in the subsidence between FLC and clear days. In fact, in this response letter we provide evidence that for most of the year, day-to-day differences in LTS over land are almost entirely driven by its surface component (T2m). The observed moisture differences clearly come from horizontal transport within the free troposphere (see also Fig. R2.7 in this document). Also, during SON, the coast-parallel winds that drive the upwelling are actually substantially weaker on FLC days than on clear days, with the SST not showing a clear pattern. To*

*summarize, multiple aspects of this proposed alternate conceptual model are not actually supported by our findings.*
*We believe, however, that here actually lies a misunderstanding: We do by no means say in this manuscript that the advection over the cool upwelling water is not a driving mechanism of Namib-region FLCs, it surely is (we also point this out on P12, L2-3)! We rather argue that the day-to-day variability that is investigated here is mainly driven by other factors – within a system in which SSTs play a key role. We now communicate this more clearly in the updated version of the manuscript as detailed below in the point-by-point responses, and have changed the title of the manuscript to: "Synoptic-scale controls of fog and low cloud variability in the Namib Desert" to more precisely describe to scope and aims of this study.*

In light of the speculative nature of the manuscript as it stands, and that the value of the climatology has already been made available to the community (in Andersen et al., 2019), I would recommend not publishing this without major revisions in order to substantiate the conceptual model of fog production presented herein.
*We would like to state that in those passages of the manuscript where we do speculate, this is clearly shown by the language used (e.g., "potentially hinting", P9 L4). We argue, however, that in light of the clear results on many aspects of the paper (which are corroborated by the additional analyses carried out for this response), the main mechanisms described in the conceptual model are well justified, offer a coherent explanation for the observed patterns and thus bring completely new insights into mechanisms that modify day-to-day FLC variability in the central Namib. As the purpose and scope of this manuscript is to better understand synoptic-scale mechanisms that influence the day-to-day variability of FLCs, very little overlap exists to the Andersen et al. (2019) paper, which 'just' provides an observation-based FLC climatology.*

Specific Comments are presented below in order of appearance:

p.1, l.6: It is not clear why these two seasons are chosen. AMJ is not a common seasonal breakdown either – it is late fall into winter. What is meant by "characterize seasonal fog" exactly?
*You are correct in pointing out that this seasonal breakdown is not common, and indeed, judging from the FLC seasonality presented in Fig. 1c), does not represent the two extreme seasons of FLC occurrence (this would be the more classical seasons DJF and MJJ/JJA). However, we have conducted this analysis in the context of understanding factors driving the day-to-day variability in fog occurrence, and in this context these seasons do make sense. During AMJ, FLCs are markedly lower in the atmosphere, leading to a maximum of fog occurrence at low-lying coastal stations ("low-FLC season" see Fig. 3 in Andersen et al. (2019)), whereas during SON, FLCs are located at higher altitudes ("high-FLC season") leading to a peak in fog occurrence at stations further inland (Andersen et al. (2019), Lancaster et al. (1984), Seely and Henschel (1998)).*
*We agree that this should be presented more clearly and have changed the corresponding sentence in the abstract to*
*"[…] during two seasons with different spatial fog occurrence patterns", and added the two following sentences to section 3.1:*
*"While the FLC occurrence in the central Namib peaks in austral summer, and is lowest during winter, due to the seasonal cycle in the vertical position of the cloud layer, fog peaks at coastal locations in AMJ and at inland locations during SON (Seely and Henschel, 1998; Andersen et al., 2019). For these reasons, this study focuses on mechanisms determining FLC variability within these two characteristic fog seasons."*

Figure 1: First a clarification - 1c) shows the average FLC occurrence over all days (from 3-9 UTC), and the peak is during the SH summertime, is that correct?

*Yes, indeed this is correct, the caption now states that "Panel c) shows monthly averages of the spatiotemporally averaged FLC cover data set."*

Also, I wonder about the wisdom of fixing this time window rigidly past the falling edge of the fog 'burn off'. Sunrise times in that area shift from ~4:00 UTC in summer to 5:45 UTC in winter, which is an appreciable portion of this 6 hr window. I worry that this could bias the FLC frequency changes observed by season.

*This is an interesting point, and indeed the time window includes the time of dissipation. In Andersen and Cermak (2018), the average diurnal cycle of FLC occurrence is shown for selected locations in the central Namib. It is apparent that diurnal FLC occurrence peaks around the time of sunrise so that within the considered time frame FLC occurrence rises, peaks and starts to dissipate. The sunrise shift potentially introducing a bias to the seasonal data sets analyzed is a good point, even though we do not analyze summer and winter so that the difference will be much smaller during the analyzed seasons. However, we are not comparing data sets of the different seasons to each other, but the difference between robustly different FLC and clear days within each season. Therefore, a marginal change in the selection of the days analyzed in each group is not expected to markedly influence the results. The choice of the time period and potential implications are now discussed in detail in section 2.1 of the updated version of the manuscript:*

*"A specified averaging time period is needed to avoid statistically mixing two separate FLC events occurring on successive nights which would be the case in a daily average FLC occurrence data set. The specific time period is chosen to include all periods of the diurnal cycle, with FLC occurrence rising, peaking, and starting to dissipate (Andersen and Cermak, 2018) during this time."*

*Later in Sec. 2.1 we state that:*

*"As the time of sunrise varies by season, the constructed data set is likely to feature a seasonal bias in FLC occurrence. It should be noted that this has no effect on the separation of FLC days and clear days within seasons, the analysis of which is the main purpose of this data set. The resulting monthly average central-Namib FLC cover (Fig. 1 c)) should not be used in a quantitative sense, but rather illustrate the general seasonal cycle of FLCs in this region."*

I think it might be useful to compare your results to any other cloud climatologies that exist for the region. For example, Dorman et al., 2019 present a COADS-based fog climatology that suggests a fog peak in MAM months in the Benguela upwelling system.

*Thank you for pointing to this interesting paper. Dorman et al. (2019) use long-term weather observations from ships to create a marine fog climatology. Marine fog patterns can be quite different from FLC patterns observed from space due to the seasonality in the vertical position of the low-cloud layer (see Andersen et al. (2019), Fig. 3 c)). The fog seasonality in Dorman et al. (2019) does not agree well with the seasonality of fog observed at coastal stations in the central Namib (Andersen et al. (2019), Fig. 3 a)), indicating that marine fog occurrence over the cool SSTs of the Benguela is not necessarily a good proxy for FLC occurrence in the central Namib. While in this paper, we look only at FLCs over land, Cermak (2012) find a maximum of marine FLCs between September and January, again highlighting the importance of the vertical position of the cloud layer when comparing satellite-observed FLCs to (marine) fog. In section 2.1 we now discuss this:*

*"It is interesting to note that the seasonal cycle of FLCs is not necessarily coupled to the seasonal cycle of fog occurrence due to the seasonal cycle in the vertical position of the low-cloud layer. For example, at coastal locations of the central Namib fog peaks between April and August (Andersen*

*et al., 2019), while marine fog over the adjacent Atlantic has been found to peak between March and May, with a minimum occurrence between June and August (Dorman et al., 2019)."*

I think the monthly FLC pattern is central enough to this work to warrant a line graph as opposed to this subtle gray scale figure which allows for a much less quantitative comparison of the seasons.
*Thank you for this comment. The visual representation of the data was done on purpose to indicate that this figure is intended to just give a description of the specific data set used in this study (which is why it is positioned in the data and methods section), and not to introduce a novel climatology. As stated above, it was not the basis for defining the seasons, and should not be interpreted in a quantitative manner, also due to the point raised by the reviewer on seasonality of the time of sunrise. We now make this clear by stating:*
*"As the time of sunrise varies by season, the constructed data set is likely to feature a seasonal bias in FLC occurrence. It should be noted that this has no effect on the separation of FLC days and clear days within seasons, the analysis of which is the main purpose of this data set. The resulting monthly average central-Namib FLC cover (Fig. 1 c)) should not be used in a quantitative sense, but rather illustrate the general seasonal cycle of FLCs in this region."*

Finally, it seems to me if you are going to carry out an annual analysis of FLC-Clear (as you do in Fig 3), you need to report what fraction of your clear and FLC days from your histogram come from each season. Because the pattern you see in Fig. 3 could match the patterns you see in Figs. 4/5 for SON simply because that is where the majority of your FLC days throughout the year come from.
*Yes this could be an issue, which is why we addressed this in the original manuscript by not just calculating the annual average difference of all FLC and clear days, but by first computing monthly average differences and subsequently averaging these. This addresses the outlined potential issues that would arise with yearly averages due to the FLC seasonality and is described on P7 L27, and also in the caption of Fig. 3 ("Averaged monthly mean differences"). We now recomputed the seasonal composites the same way to also address within-season changes in FLC occurrence.*

p.7, l.15: This is confusing because you are focusing on SON, and only the thin latitudinal band from ~22-24°S, the FLC peak actually occurs in DJF (as shown in Fig. 1c & Andersen 2019, Fig. 2c.)
*We agree that this could be communicated more clearly. As outlined in our first response, the seasons were chosen with fog patterns in mind. The fog seasonality is related to FLC characteristics (vertical position), and, in Andersen et al. (2019), this lead to the definition of the two seasons that we investigate with this paper (SON and AMJ). We now state in this paragraph that:*
*"While the FLC occurrence in the central Namib peaks in austral summer, and is lowest during winter, fog peaks at coastal locations in AMJ and at inland locations during SON due to the seasonal cycle in the vertical position of the cloud layer (Seely and Henschel, 1998; Andersen et al., 2019). For these reasons, this study focuses on mechanisms determining FLC variability within these two seasons. "*
*The study region in the central Namib (~22-24°S) is chosen because nearly all research on Namib-region fog is conducted in this region, as the historical and current station measurements stem from here (Nagel 1969, Nieman et al. 1978, Lancaster et al. 1984, Seely and Henschel 1998, Henschel and Seely 2008, Kaseke et al. 2017, 2018, Li et al. 2018, Spirig et al. 2019, Wang et al. 2019). We now mention some of these sources in section 2.1 for added clarity.*

p.8, l.3: The winds are southerly throughout the region, how do you infer "northerly" advection?
*While indeed, over the ocean the winds are southerly, over land this is not the case. On P8 L3, we refer to the T2m anomaly that is most pronounced in the Kalahari region. In this region, northerly*

*advection is quite common (see e.g. Fig. 2b)), and Fig. 3a) shows that the northerly wind component into this region is strengthened on FLC days. To more clearly illustrate this, Fig. R2.1 in this response shows seasonal average T2m and 10m winds for FLC and clear days. The figure shows considerable temperature differences over land (specifically in the Kalahari region) and also the associated northerly winds that transport heat into the region. While during AMJ, the northerly flow in the Kalahari is also apparent on clear days, it does not originate from warmer regions, i.e. no substantial heat transport is expected. This figure is now added to the appendix of the paper.*

[Figure]

*Fig. R2.1: Seasonal average ERA5 T2m and 10m winds in SON (left-hand panels) and AMJ (right-hand panels) for FLC (top) and clear (bottom) days. Winds are averaged to a 1°x1° resolution, the considered time period is 2004-2017.*

p.8, l.7: You are referring to features of the climatological Z500 pattern without showing what that is, so it is hard to assess these statements about a trough and the absence of a coastal low. Are you sure Olivier and Stockton (1989) are not referring to a particular time of year for their coastal trough as opposed to a year round analysis that you are presenting here? A quick look at NCEP reanalysis data for the region shows a subtle trough upwind of the coastline.

[Figure]

*Thank you for this comment, the sentence on the coastal low was not communicated clearly enough. While one can see lower pressure along the coastline in the climatology of MSLP in Fig. 2 a) and b) of the manuscript, we do not find any indication for an increase in the occurrence of a coastal low on FLC days. There might still be coastal lows, of course, which is discussed on P8 L9-10. While Olivier and Stockton (1989) conduct a yearly analysis, the study does provide seasonal details that we agree not to have discussed enough. They find that the linkage between the coastal low and FLC occurrence in Lüderitz is most pronounced between November and March, and much less so from June to August, when cold fronts are more often associated with FLCs in Lüderitz. We would like to point out, though, that Olivier and Stockton (1989, p.73) state that "The coastal low which causes fog is a relatively small, local phenomenon". Therefore, while FLC occurrence in Lüderitz may be driven by the coastal low, this may not be directly related to FLCs in the central Namib as defined in this paper. This is also visualized clearly in Haensler et al. 2011 (Fig. 1a)). In fact, in Lüderitz, FLCs are much less common than in the central Namib (Fig. 2b) and c) in Andersen et al. (2019), latitude of Lüderitz: ~26.6°S), indicating that these are likely two separate regimes. Another interesting point that needs to be included in the manuscript is that more upwelling does not necessarily lead to increased FLC occurrence in the Namib region, as Olivier and Stockton (1989) point out that in the case of Lüderitz, an upwelling extent of greater than 200km actually leads to less FLCs in Lüderitz, as the local phenomenon of the coastal low is not able to transport moist air from beyond the upwelling front.*

*We now provide additional details in the introduction:*

*"In Olivier and Stockton (1989), a coastal low is described as the mechanism that, in case of a narrow coastal upwelling region, drives the onshore advection of foggy air masses into the region of Lüderitz in southern Namibia during austral summer, while during winter they find fog to be associated with cold fronts. However, they assume that, while undetected, coastal lows were also present in these cases, as they typically precede the passage of a cold front (Olivier and Stockton, 1989; Reason and Jury, 1990)."*

*Concerning the climatology of the region, described in Sec. 3.1 we now state:*

*"Coastal upwelling, which has been shown to determine marine sea fog patterns along the Namibian coastline (Dorman et al., 2019), in combination with the presence of a coastal low that drives the onshore advection of foggy air masses have been found to be major drivers of fog occurrence in southern Namibia during austral summer (Olivier and Stockton, 1989). One should note though that the relationship between SSTs and Namib-region fog is complex, as Olivier and Stockton (1989) point out that a too large upwelling extent can also lead to less fog in southern Namibia. Based on these insights, and also on knowledge from related coastal upwelling systems (Cereceda et al., 2008; Johnstone and Dawson, 2010; Del Río et al., 2018; Dorman et al., 2019), it*

*is clear that the Atlantic anticyclone, the SSTs, and the large-scale subsidence are main drivers of this coastal FLC system."*

*Concerning the seasonal differences described in section 3.2 we now state that*

*"While a coastal low, which has been described in Olivier and Stockton (1989) as a local feature that can determine onshore flow, may still be present on FLC days, there is no indication of an increase in its presence on FLC days on average. However, as Reason and Jury (1990) describe, the coastal low is frequently followed by a frontal passage, which is a synoptic-scale signal observed here (Fig. A1)."*

*The average Z500 patterns are now included as additional contours in Fig. 2 of the original manuscript.*

p.9, l.4: I think this SST time lag inference is unfounded speculation on the authors' part. The wind difference indicates to me that the clear days have slightly stronger offshore wind components, which could weaken ocean upwelling. It is the alongshore wind component that determines the upwelling, and could possibly have subtle variations due to coastline geography (see, for example, Koračin, Darko, Clive E. Dorman, and Edward P. Dever. "Coastal perturbations of marine-layer winds, wind stress, and wind stress curl along California and Baja California in June 1999." Journal of Physical Oceanography 34, no. 5 (2004): 1152-1173.)

*We agree that this specific statement is somewhat speculative, which is expressed by our cautious phrasing in the manuscript ("potentially hinting"). Indeed, clear days feature offshore winds, which is especially pronounced during AMJ. During this time of the year, we agree that upwelling may be more intense on FLC days, as the alongshore wind component is stronger. However, during SON we find that the alongshore wind that drives the upwelling is substantially weaker on FLC days than on clear days (see Fig. R2.2 of this document, top row). This would mean that upwelling should be reduced on FLC days, but SSTs do not show significant differences. However, in the context of interpreting these wind-ocean interactions, it is important to note the role of time scale, as the Ekman transport is not only dependent on the instantaneous wind field, but produces a steady-state situation only after a few pendulum days (Pond and Pickard, 2013). As such, the upwelling reacts to time-integrated winds, introducing complex time lag effects, which are investigated in e.g. Goubanova et al. (2013). We now discuss this in the updated version of the manuscript by stating:*

*"[…] potentially hinting at a time-lag response of SSTs, which is to be expected, as Ekman transport produces a steady-state situation only after a few pendulum days (Pond and Pickard, 2013)."*

*We also discuss coastal modulation of winds on the basis of the suggested publication in section 3.1:*

*"On a local scale, the near-coastal winds that drive the upwelling are additionally modulated by the coastal topography (Koračin et al., 2004)."*

[Figure]

*Fig. R2.2: Mean of monthly averaged MSLP and 10m winds during SON (top) and AMJ (bottom) on FLC days (left-hand panels), clear days (center), and their difference (right-hand panels).*

p.9, l.5: The SST anomaly having a hydrostatic impact on MSLP seems highly unlikely given that the FLC effects are associated with strong synoptic forcing as argued in the last paragraph. Furthermore, how exactly does it appear likely that SST-FLC correlations are most 'pronounced on seasonal scales'? Can't that be determined for your data set and put to the test? There is not all that much variability in SST in this region, as far as I can see from NCEP reanalysis data.

*Thank you for this comment. We developed the hypothesis of the hydrostatic impact on MSLP on the basis of the yearly differences shown in Fig 3 a) of the manuscript. After consideration of your comment, we delete this hypothesis from the manuscript, as it is not corroborated by the seasonal patterns of Fig. 4 of the manuscript.*

*About the comment on SST seasonality and variability: Klein and Hartmann (1993) have shown that the seasonal cycle of SSTs dominates the seasonality in LTS (Fig. R2.3 of this response) in the Namibian stratocumulus field. Hutchings et al. (2009) state in their abstract that:"The southern Benguela region is characterised by a pulsed, seasonal, wind-driven upwelling at discrete centres [...]", but Tim et al. (2015) state on page 484 that "[...] a clear picture of the upwelling seasonality is not established yet." Concerning other timescales of SST variability, Goubanova et al. (2013) note that subseasonal SSTs close to the coast feature two regimes of variability: an 11 day oscillation, and a 61 day oscillation. In the updated version of the manuscript, we now state that "It appears likely that effects of SST patterns on FLC variability are most pronounced on time scales (i.e. seasonal to interannual) that feature higher SST variability (Hutchings et al., 2009; Goubanova et al., 2013; Tim et al., 2015), as also observed in the Chilean Atacama desert (Del Río et al., 2018)."*

*We agree that it would be useful to further explore SST influence on longer time scales, but 14 years might not be enough for meaningful statistical analyses of e.g. seasonal relationships. Also, with this paper, we are specifically looking at day-to-day variability within these seasons.*

[Figure]

*Fig. R2.3: The seasonal cycle of the SST and 700 mb Temp components of LTS in the Namibian stratocumulus clouds field (Figure taken from Klein and Hartmann (1993)).*

p.9, l.10: This speculation would benefit from some sort of simple calculation of the magnitude of this effect. Are you meaning to say that radiative cooling will be significantly influencing the SST's? If so, this seems unlikely in a strong upwelling system such as this. Or are you saying that FLC, once formed, will be sustained by effective cloud-top radiative cooling due to the dry tongue over it? As it stands this just seems like a qualitative speculation that is unsubstantiated (without at least a reference to another work that has explored a comparable situation, or a back of the 'envelope' calculation on your part.) The same holds for the assertion that moisture advection influences the surface heat low by principally radiative means presented in the following sentences. *While the TCWV differences might not seem to be much in absolute terms (between ~2 and 5 kg m$^{-2}$), in relative terms, they are quite substantial. Also, in case of the free-tropospheric dry anomaly over the coast, one should note that part of the dry anomaly is actually masked by the moist anomaly within the marine-boundary layer (see Fig. 6 in the manuscript). As we point out on page 11, lines 26-29, the relative difference in Q in the dry anomaly is as high as 220%, and in the continental moist anomaly, Q is about twice as high on FLC days than on clear days. Over the land this will certainly lead to a substantial warming, as observed during AMJ, and has been observed in the Kalahari (Manatsa and Reason (2017)), and in other dry subtropical deserts before (e.g. the Sahara: (Evan et al. 2015, Alamirew et al. 2018), or the Sahel (Oueslati et al. 2017)). In Fig. R2.4 of this response, we show the statistical relationship between the TCWV anomalies and the T2m anomalies during AMJ over land for pixels which feature significantly higher T2m on FLC days than on clear days (cf. spatial patterns of Fig. 5 b) and d) of the original manuscript). A clear relationship is obvious, with a Pearson correlation coefficient of 0.75. This statistical relationship indicates that more than half of the observed T2m anomalies can be statistically explained by the TCWV anomalies, underscoring the relevance of the greenhouse effect for T2m.*
*In section 3.2 of the updated version of the manuscript we now state that*
*"These moist air masses may contribute to the observed T2m heat anomaly via greenhouse warming (Fig. 3 c)). This effect of free-tropospheric moisture on surface temperatures has been observed in the Kalahari (Manatsa and Reason, 2017)  and other arid or semi-arid regions before (Evan et al., 2015; Manatsa and Reason, 2017; Oueslati et al., 2017; Alamirew et al., 2018)."*
*In section 3.3 of the updated version of the manuscript we now discuss this in more detail:*
*"Here the T2m anomalies closely follow those of the TCWV (Pearson correlation coefficient of 0.75 in continental regions with significantly higher T2m on FLC days than clear days), suggesting that*

*the increased moisture causes an additional surface heating due to greenhouse warming as discussed in Sec. 3.2. It is likely that the TCWV anomaly is caused by a large-scale free-tropospheric moisture transport from the tropics, which is supported by the marked wind anomalies at 500hPa (Fig. 4 d)) that show a northwesterly anomaly, and the absolute wind and moisture fields at 700 hPa during this time (Fig. A1)."*

[Figure]

*Fig. R2.4: The relationship of TCWV anomalies and T2m anomalies over land during AMJ, and in regions where T2m is significantly higher on FLC days than on clear days. The figure is added to the appendix of the manuscript.*

*While the SST will surely not respond as strongly as the LST, and in fact might be negligible, studies exist that point to the impact on stratocumulus clouds in the southeastern Atlantic. For this region, Adebiyi et al. (2015, p. 2015) have shown that with an increase in "midtropospheric moisture of about 1.2 g kg$^{-1}$, the downwelling longwave radiation averaged between 550 and 750 hPa increases by about 15 Wm$^{-2}$, reducing the net longwave cloud-top cooling by the same amount[...]". Recently, Adebiyi and Zuidema (2018) have shown that in the southeastern Atlantic, free-tropospheric moisture has a significant effect on stratocumulus cloud cover, where increases in free-tropospheric moisture are associated with a decrease in low-cloud cover. With specifically chosen variations of the predictors selected in their multivariate statistical model they could attribute this cloud response clearly to the greenhouse effect of water vapor. It is interesting to note that during the summer season (DJF), the time of maximum FLC occurrence, this dry anomaly is far larger, with anomalies up to 12 kg m$^{-2}$ as shown in Fig. R2.5 of this document.*

[Figure]

*Fig. R2.5: left: DJF average TCWV; right: mean of monthly average TCWV differences (FLC days - clear days) during DJF.*

*Throughout the manuscript we have changed the wording from "facilitating FLC formation" to "increasing FLC cover".*
*In section 3.2 of the updated version of the manuscript, this is discussed in more detail:*
*"This is likely the dry slot (Browning, 1997) or dry air intrusion of the synoptic-scale disturbance that leads to increased longwave cooling at cloud top in case of FLC presence, which has been shown to be a main determinant of cooling within the marine boundary layer (Koračin et a. 2005). This enhanced cooling can increase FLC cover, which has been observed to be a significant mechanism for stratocumulus clouds over the southeastern Atlantic (Adebiyi and Zuidema, 2015; Adebiyi et al., 2018)."*
*In section 3.3 we state that*
*"It is interesting to note that the marine dry anomaly peaks between December and February (not shown), the season with maximum FLC cover in the central Namib, with TCWV anomalies exceeding 10 kg m$^{-2}$."*

p.9, l.14/15: This hypothesis could be tested by looking at the T anomaly only during the overnight hours to see if it is an air mass difference or an insolation difference (I strongly suspect it is the latter.) My hunch is that it will be slightly warmer overnight because of radiative heating of the surface from the FLC, which would provide evidence against the air mass difference hypothesis.
*We have conducted the suggested analysis and find that during nighttime (1 UTC and 3 UTC), the coastal regions are significantly cooler, pointing to a difference in air mass between FLC and clear days (cf. Fig R2.6 of this response letter for 1 UTC differences). We discuss now this in the manuscript:*
*"As this anomaly is also apparent during nighttime (1 and 3 UTC, not shown), it is likely that this pattern is mainly due to the relatively warm subsiding continental outflow that is apparent on clear days, rather than a radiative effect of FLCs as found in California (Iacobellis and Cayan, 2013)"*

[Figure]

*Fig R2.6: Averaged monthly mean differences (FLC days - clear days) of T2m at 1 UTC. In each pixel, an independent two-sided t-test is computed to identify significant differences between FLC and clear days for each month. Contours mark regions where the distributions differ significantly at the 0.01 level (median of the monthly p values <0.01).*

*We also discuss the correlation between T2m and coastal low cloudiness found in Clemesha et al (2017):*
*"It should be noted that in a comparable upwelling system (coastal California), Clemesha et al. (2017) also find a positive relationship between T2m over land and coastal low-level cloudiness, with the T2m anomaly shifted poleward by about 5° latitude from the cloud field. They propose that the T2m-cloud relationship is due to spatially-offset associations between coastal low-level cloudiness and stability (potential temperature at 700 hPa), which is strongly correlated to T2m over land thereby resulting in the T2m anomaly, rather than T2m driving the onshore advection. While in the central Namib, the anomaly patterns between potential temperature at 700 hPa and T2m are similar in that they are also positively correlated during SON (and therefore compensate each other in terms of LTS, Fig. 5 c) and e)), they are uncorrelated during AMJ (and also in the annual averages), when T2m over land is strongly correlated to TCWV. Also, during all times of year, the T2m and MSLP anomalies are directly inland from the cloud field, suggesting an influence on onshore advection."*

p.9, l.27: In the discussion surrounding the similar annual pattern of Fig. 3b you referred to it as a trough instead of a cut-off low.
*After reviewing the absolute fields of Z500 and Z700, we conclude that the synoptic-scale disturbances are not strictly speaking cut-off lows and Rossby wave breaking, and have deleted the corresponding wordings. The trough is visible in the absolute wind fields shown in Fig. R2.7 of this document. This figure is now included in the appendix of the updated version of the manuscript.*

p.11, l.4: A few 0.1's K is a subtle change, but the increased wind speeds could definitely increase the latent heat fluxes in the upwind region. Here, you could get a sense of the relative magnitude of these effects by using a simple moisture exchange coefficient and quantifying differences in saturation vapor pressures vs. mean wind speeds.
*Yes, this increase in upwind latent heat fluxes leading to increased marine-boundary layer moisture is precisely our hypothesis. However, we agree with Reviewer 2 that we did not state this clearly enough. We do not believe that quantifying the contribution of each specific mechanism is within the scope of this manuscript, and particularly in this case, as we would also need to take into account horizontal transport and vertical mixing. We state more clearly now:*

*"In isolated patches further south, upwind of the study area, SSTs tend to be significantly higher on FLC days. This could lead to increased surface latent heat fluxes, increasing the moisture content of the marine boundary layer, particularly during AMJ when stronger near-surface winds are also apparent. A few 100 km to the west and south of the Namibian coastline, SSTs could similarly add to the increased moisture within the marine boundary layer."*

p.11, l.6: I would bet that it has everything to do with upwelling induced by the wind field.
*This aspects seems to be dependent on the considered time scale. We now state that:*
*"It is not clear yet, however, what exactly drives the observed anomaly patterns of SSTs. As upwelling reacts to the time-integrated wind field forcing over longer time scales than analyzed here (Pond and Pickard, 2013), the SST response to the instantaneous winds that are considered here is expected to be relatively weak. However, in the case of a relatively stationary disturbance as discussed above, the upwelling patterns could indeed reflect an SST response to a synoptic forcing."*

p.11, l.30: Or the dry anomalies could be associated with subsidence which augments the LTS in the fog cases. This reduces MBL entrainment and along with increased LH fluxes upwind helps to build up Q in the MBL. Along with a lower SST, these influences act in tandem to reduce the dew point depression.
*In Fig. 6 we do not find substantial differences in subsidence, and during SON the opposite is actually the case (the increased stability over the ocean is driven by the decrease in T2m, likely due to relatively warm continental outflow on clear days). In Fig. R2.7 of this document, we show the average Q and winds at 700 hPa, which is the layer with strongest Q differences (see Fig. 6 of the manuscript). It is clearly apparent that the dry anomaly during both seasons and the moist anomaly over continental Africa during AMJ is driven by horizontal transport induced by the synoptic disturbance on FLC days. We agree that the increased LH fluxes upwind should build up Q in the MBL, which we state on P11 L4. In the original version of the manuscript, we did not clearly enough present the link between increased winds, SST and MBL Q. This is now clarified in the updated version of the manuscript:*
*"In isolated patches further south, upwind of the study area, SSTs tend to be significantly higher on FLC days. This could lead to increased surface latent heat fluxes, increasing the moisture content of the marine boundary layer, at least during AMJ when stronger near-surface winds are apparent. A few 100 km to the west and south of the Namibian coastline, SSTs could similarly add to the increased moisture within the marine boundary layer."*

[Figure]

*Fig. R2.7: Seasonal averages of Q and winds at 700hPa on FLC days (left), clear days (center), and their difference (right) during AMJ (top) and SON (bottom).*

Figure 5: Very little attention is paid to the LTS anomalies presented. My read of Figs. 4/5 is that regardless of season FLC is strongly associated with low SSTs, low T2m, and high LTS.
*In general, we agree that FLCs will preferentially occur in situations with low SSTs and high LTS. This is actually the reason to show the climatology in Fig. 2 of the manuscript. Of course, the region is rich in FLCs due the low SSTs and the subsidence that combine to lead to very stable conditions. We actually state that "these stable conditions promote the formation of the southeastern Atlantic stratocumulus cloud deck and determine its seasonal cycle [...]". Also, Fig. 3 shows that the differences in LTS over land are almost entirely caused by the surface component of LTS (Pearson correlation coefficient is -0.90 for land pixels for the yearly averages). During SON, this relationship is weaker, which is now discussed in the updated version of the manuscript. We now describe seasonal differences in LTS in a new paragraph:*
*"While the yearly averaged composites show that over land, LTS is driven to a large extent by T2m (Fig. 3 c) and d)), this is not quite as pronounced during SON (correlation coefficient =-0.57; Fig. 5 c) and e)). Over continental southern Africa, the differences in T2m (Fig. 5 c)) are frequently compensated by similar differences in potential temperature at 700 hPa (not shown). The most pronounced LTS feature during both seasons, however, is the coastal anomaly of increased LTS (over land and weaker over the adjacent ocean), which is driven by T2m. As this anomaly is also apparent during nighttime (1 and 3 UTC, not shown), it is likely that this pattern is mainly due to the relatively warm subsiding continental outflow on clear days, rather than a radiative effect of FLCs as found in California (Iacobellis and Cayan, 2013). During AMJ, LTS is significantly lower over a large marine region south of 25°S, which is likely caused by the synoptic-scale disturbance."*

p.14, l.17: You could look at potential temperature to see what sort of effects that radiative cooling has on the foggy days. It seems that potential temperature would be a better variable to present in the back trajectories (unless, of course, it is a purely isentropic back trajectory.)
*In the updated version of the manuscript, the potential temperature is additionally shown in the backtrajectory figure. It suggests that the main difference between the trajectory groups remains to be the difference in MBL Q, as stated in the original version of the manuscript.*

p.14, l.21 to p.15, l.1: Doesn't this contradict your hypothesis presented earlier about the lower column water vapor leading to greater radiative cooling on the foggy days?

*No, as in Fig. 8, different subsets of the data are compared (only the blue lines of Fig. 7 a) and c)), and differences are computed by following the backtrajectory. As shown in Fig. 5 a) of the manuscript, the lower TCWV is mostly a local phenomenon that is especially pronounced north of ~23°S, and would therefore modify the trajectories only for a limited time.*

p.18, l.19: It is not too surprising that so much is explained by the MSLP fields because they determine a lot of things. For instance, MSLP is the main variable used in calculating conventional upwelling indices. Again, I found the lack of centrality of coastal SSTs to be surprising in this work given how important it is found to be in most other studies.

*Yes, indeed changes in MSLP can modify upwelling intensity. However, we would argue that to first order, the differences in MSLP that we find in this paper explain most of the variability in FLC occurrence because they comprise the information on the marked differences in dynamics between FLC and clear days which is clearly shown in the contrasting backtrajectories of Fig. 7 of the manuscript. This is the first-order mechanism, as the offshore winds that are apparent on most clear days (Fig. 7 of the manuscript and Fig. R2.2 of this document) will hinder onshore advection of moist/cloudy marine-boundary layer air masses. In that case it does not matter for FLC occurrence in the Namib whether marine fog or low clouds are formed or not. We do agree that SSTs could be included in the discussion of these results, and will likely be relevant for marine fog occurrence as analyzed in e.g., Dorman et al. (2019).*

*The manuscript now discusses this in more detail:*

[revised manuscript text omitted]

---

## Author Comment (AC3) · 7 Jan 2020

**Synoptic-scale controls of fog and low clouds in the Namib Desert: Response to Reviewer 3**

**Hendrik Andersen, Jan Cermak, Julia Fuchs, Peter Knippertz, Marco Gaetani, Julian Quinting, Sebastian Sippel, and Roland Vogt**

contact: **hendrik.andersen@kit.edu**

*We would like to thank reviewer 3 for her/his review of the manuscript and the valuable comments. Comments by the referee are colored in black, our replies or comments are colored in blue and written in italics.*

This is a well-written article. The authors are to be congratulated.

However, It has long been known that fog and low cloud in the coastal zone of the Namib and the South African west coast are largely due to a local/meso-scale phenomenon called a coastal low. This is a weak low pressure system trapped between the western escarpment to the east and the Benguela current to the east. It only extends to just above the height of the escarpment. The diameter of the coastal low and the extent of the cold water upwelling region often determines whether fog occurs or not. An interplay between an approaching cold front and a HIGH pressure system over the continent is thought to cause the coastal low and associated fog to move southwards from the Namibian coast, down the South African west coast, around the tip of South Africa and northwards towards Kwazulu-Natal. It is unclear why the authors need to work at synoptic scale when the phenomenon occurs at a much smaller scale. The role of a cut-off low in fog occurrence is really surprising.

*We agree with referee 3 that the connection between coastal lows and fog occurrence has been suggested for a long time, to our knowledge the main paper on this is titled "The influence of upwelling extent upon fog incidence at Lüderitz, southern Africa" by Olivier and Stockton (1989), which is discussed on pages 7 and 8 of the original manuscript. In this very interesting study, on the basis of two years (1983 & '84) of satellite observations, Olivier and Stockton find that fog occurrence in Lüderitz is mostly associated with coastal lows, especially in austral summer, while during winter, it is associated with cold fronts. They go on to assume that as coastal lows often precede the passage of cold fronts (observed to be associated with fog during winter), a coastal low "was present, but unobserved, during these conditions." (p. 71). While their paper is focused on fog occurrence in Lüderitz, southern Namibia, this concept is extended to other regions along the southern African coast, based on two conference papers (Estie 1984, Sciocatti 1984). Both of these publications could not be found in the usual online publication data bases and are therefore not cited in the manuscript.*

*However, other hypotheses also exist:*

1. *Seely and Henschel (1998) suspect that the onshore advection of 'high fog' is enhanced by the plain-mountain wind.*

*2. Lancaster et al. (1984) hypothesize that the seasonal occurrence patterns of fog in the central Namib is influenced by the continental high pressure system, pointing to a synoptic-scale influence.*

*3. In the last few years, several papers have been published that question the described mechanism, and typify most fog events as locally generated radiative fog (e.g., Kaseke et al. 2017, 2018).*

*Therefore, we believe that substantial knowledge gaps do still exist and that good reasons exist to study the mechanisms driving variability of fog and low clouds along the south western African coastline on the basis of an extensive observational data set as done here. Also, many of the anomaly patterns that we find are actually on synoptic scales (e.g., moisture transport), underscoring the relevance of this scale to gain a more complete understanding of mechanisms influencing the Namib-region FLC system.*

*The note of the cut-off low, however, was a mistake on our part, and is now corrected in the updated version of the manuscript.*

It is suggested that much more information is provided on the research that has already been conducted on the occurrence of fog along the southern African west coast.

*We thank reviewer 3 for this useful suggestion. We have now added a much more detailed discussion on coastal lows in different sections of the manuscript.*
*In the introduction:*
*" In Olivier and Stockton (1989), a coastal low is described as the mechanism that, in case of a narrow coastal upwelling region, drives the onshore advection of foggy air masses into the region of Lüderitz in southern Namibia during austral summer, while during winter they find fog to be associated with cold fronts. However, they assume that, while undetected, coastal lows were also present in the case of cold fronts, as they typically precede the passage of a cold front (Olivier and Stockton, 1989; Reason and Jury, 1990)."*
*In section 3.1:*
*"Coastal upwelling, which has been shown to determine marine sea fog patterns along the Namibian coastline (Dorman et al., 2019), in combination with the presence of a coastal low that drives the onshore advection of foggy air masses have been found to be major drivers of fog occurrence in southern Namibia during austral summer (Olivier and Stockton, 1989). One should note though that the relationship between SSTs and Namib-region fog is complex, as Olivier and Stockton (1989) point out that a too large upwelling extent can also lead to less fog in southern Namibia."*
*In section 3.2:*
*"While a coastal low that has been described in Olivier and Stockton (1989) as a local feature that can determine onshore flow may still be present on FLC days, at least in some of the cases, the composite differences between FLC days and clear days do not provide a clear indication of an increase in its presence on FLC days. However, as Reason and Jury (1990) describe, the coastal low is frequently followed by a frontal passage, which is a synoptic-scale signal observed here (Fig. A1)."*

*We believe to now extensively discuss the existing literature on fog along the southern African west coast, and in the updated version of the manuscript also discuss in much more detail links to comparable upwelling systems, but if reviewer 3 knows of an important paper on the regional mechanisms that is still missing, we would kindly ask her/him to point us to this publication.*

*References*

*Dorman, C. E., Mejia, J., Koracin, D., and McEvoy, D.: World marine fog analysis based on 58-years of ship observations, International Journal of Climatology, pp. 1–24, https://doi.org/10.1002/joc.6200, 2019.*

*Estie, K. E.: 'Forecasting the formation and movement of coastal lows', Paper presented at the Coastal Low Workshop held at the Institute of Marine Technology, Simonstown, 15 and 16 March 1984, 17-27, 1984.*

*Kaseke, K. F., Wang, L., and Seely, M. K.: Nonrainfall water origins and formation mechanisms, Science Advances, 3, e1603 131, https://doi.org/10.1126/sciadv.1603131, http://advances.sciencemag.org/lookup/doi/10.1126/sciadv.1603131, 2017.*

*Kaseke, K. F., Tian, C., Wang, L., Seely, M., Vogt, R., Wassenaar, T., and Mushi, R.: Fog spatial distributions over the central Namib Desert - An isotope approach, Aerosol and Air Quality Research, 18, 49–61, https://doi.org/10.4209/aaqr.2017.01.0062, 2018.*

*Lancaster, J., Lancaster, N., and Seely, M. K.: Climate of the central Namib desert, Madoqua, 14, 5–61, 1984.*

*Olivier, J. and Stockton, P. L.: The influence of upwelling extent upon fog incidence at Lüderitz, southern Africa, International Journal of Climatology, 9, 69–75, https://doi.org/10.1002/joc.3370090106, 1989.*

*Reason, C. J. and Jury, M. R.: On the generation and propagation of the southern African coastal low, Quarterly Journal of the Royal Meteorological Society, 116, 1133–1151, https://doi.org/10.1002/qj.49711649507, 1990.*

*Sciocatti, B.: The effect of sea surface temperatures on the formation or disposal of advection fog with the passage of a coastal low, Paper presented at the Coastal Low workshop, held at the Institute of Marine Technology, Simonstown, 15 and 16 March 1984, 51–52, 1984.*

*Seely, M. K. and Henschel, J. R.: The Climatology of Namib Fog, Proceedings, First International Conference on Fog and Fog Collection, Vancouver, Canada, pp. 353–356, 1998.*

---

## Referee Report (RR1)

Reviewer 2
Response to authors' rebuttal of review of "Synoptic-scale controls of fog and low cloud variability in the Namib Desert"

Overall I found the authors response to my initial comments quite thoughtful and mostly satisfactory. I now recommend that the revised manuscript be published in Atmospheric Chemistry & Physics without the need for further review by me. Nevertheless, in the spirit of continued dialogue on this fascinating topic I present below my final responses to their responses in the event that they might find them useful in making any final improvements to this or future manuscripts, to be done at their own discretion.

About the change of title to focus on 'variability' of fog/low-cloud (FLC) instead of just FLC: I do not see a substantive difference in describing what controls the presence of fog/low-cloud and its *variability*. In order to understand the variability you need to understand the causes, and presumably knowing the dominant factors of causation will tell you something about the variability. I do not understand the distinction being made here. The discussion spans several temporal scales and those distinctions seem more critical than the one between FLC and its *variability*.

About the shift in sunrise times across the year: It is too bad not to treat Figure 1c as quantitative because it would be nice to have a quantitative assessment of FLC cover across the annual cycle in the region for reference. I suppose you are suggesting that this is presented in Fig. 3a of Andersen et al. (2019). It is your manuscript, but I don't see why you don't use a window that follows the sun, or is pushed back off the falling edge of the stratocumulus deck in order to account for this periodicity methodically.

About the original comment surrounding p.7/l.15: OK. I see now that I believe some of the confusion arose from the use of 'high' and 'low' which could refer to frequency or elevation. Being more specific with the language in your revision helps a lot.

About Fig. R2.1: Yes, I see your point now. I was misinterpreting Figure 2, possibly because the warm colors of the MSLP gave me the impression of temperature. My apologies.
This supplemental figure shows N-NE advection into the inland region. You might consider contouring the T-field so that advection is more easily seen. Also, you do not present any wind speed legend for the vector lengths so it is difficult for the reader to estimate an advection rate.

About the discussion of SST-FLC correlations on seasonal scales and your inclusion of Klein & Hartmann (1993) Fig. 6a in your response: You cite their showing SST dominating annual LTS changes, but you are arguing that LTS does *not* influence your day-to-day variability of FLC in your domain. Furthermore, the seasonal frequency of stratus from Klein & Hartmann (1993), with its distinct peak in SON, does *not* match your data set (their Fig. 4b). Perhaps that is because theirs is for 10-20°S, 0-10°E, a region downwind of yours.

About the original comments surrounding p.9/l.10: I appreciate the extensive revisions you included in the discussion of TCWV, especially the references that attempt to quantify the longwave radiative impacts of the drier lower troposphere. However, I would suggest that your arguments with correlation should be taken with a grain of salt as they are by no means indicative of causation. This is particularly true in the case of water vapor and temperature, two meteorological variables that are highly correlated for strictly thermodynamic, and non-radiative, reasons. For example, I would not be surprised if a large portion of your Fig. R2.4 were not merely a manifestation of the Clausius-Clapeyron relationship.

About Figure R2.7: I appreciate the addition of the 3-panel (FLC, and clear, and difference) format of this new supplemental figure.

About upwelling winds and SST response: Just to put a finer point on this SST lag to upwelling winds, from experience I expect that the lag time is something of order 5-15 hours in the midlatitudes. This is more like the period of an inertial oscillation, which is half of a pendulum day. In other words, I do not think you need to wait around for an equilibrium solution time scale of a couple of pendulum days to get the bulk of the temperature response to a wind shift. It is observed to occur much more rapidly. [See for example, Lentz, S.J., J. Phys. Ocean., 1992]

About your subsidence reported in Figure 6: In regions of complex terrain and coastal geography I suspect that the low spatial resolution in the ERA5 reanalysis data probably does not capture subsidence very completely, but I realize it is the best you have available, and your other arguments are persuasive.

About your inclusion of potential temperature in Figure 8: I disagree that the primary difference is definitely the MBL Q (1.2-1.5 g/kg). The difference in potential temperature between clear and FLC days is ~2.5 K. At those temperatures the dQ/dT is about 0.7 g/kg/K, so those differences are comparable with respect to dew point depression.

In conclusion, I would like to thank the authors for their thorough and thoughtful responses to my review and for putting the significant effort into advancing the conversation and my understanding of fog and low cloud in these upwelling systems. Excellent job!

---

## Author Response (AR2)

**Synoptic-scale controls of fog and low clouds in the Namib Desert: Response to Reviewer 2**

**Hendrik Andersen, Jan Cermak, Julia Fuchs, Peter Knippertz, Marco Gaetani, Julian Quinting, Sebastian Sippel, and Roland Vogt**

contact: **hendrik.andersen@kit.edu**

*We would like to thank reviewer 2 for her/his 2nd review of the manuscript, the valuable comments, and the continued discussion. Comments by the referee are colored in black, our replies or comments are colored in blue and written in italics.*

Overall I found the authors response to my initial comments quite thoughtful and mostly satisfactory. I now recommend that the revised manuscript be published in Atmospheric Chemistry & Physics without the need for further review by me. Nevertheless, in the spirit of continued dialogue on this fascinating topic I present below my final responses to their responses in the event that they might find them useful in making any final improvements to this or future manuscripts, to be done at their own discretion.
*Thank you for the positive evaluation. We appreciate the continued dialogue.*

About the change of title to focus on 'variability' of fog/low-cloud (FLC) instead of just FLC: I do not see a substantive difference in describing what controls the presence of fog/low-cloud and its \*variability\*. In order to understand the variability you need to understand the causes, and presumably knowing the dominant factors of causation will tell you something about the variability. I do not understand the distinction being made here. The discussion spans several temporal scales and those distinctions seem more critical than the one between FLC and its \*variability\*.
*In general, we agree that there can be similarities between the drivers of a system and the drivers of a system's variability. In this case, however, the SSTs, while being a main characteristic feature of the Namibian FLC system, do not seem to be a main driver for its day-to-day variability. Hence we believe that the clearer language in the title and the text improves the manuscript.*

About the shift in sunrise times across the year: It is too bad not to treat Figure 1c as quantitative because it would be nice to have a quantitative assessment of FLC cover across the annual cycle in the region for reference. I suppose you are suggesting that this is presented in Fig. 3a of Andersen et al. (2019). It is your manuscript, but I don't see why you don't use a window that follows the sun, or is pushed back off the falling edge of the stratocumulus deck in order to account for this periodicity methodically.
*Yes, this is presented in the paper Andersen et al. (2019). We think that this 'easy' way of averaging the data helps the readability of the manuscript and therefore decided to leave it in its current form.*

About the original comment surrounding p.7/l.15: OK. I see now that I believe some of the confusion arose from the use of 'high' and 'low' which could refer to frequency or elevation. Being more specific with the language in your revision helps a lot.
*Ok!*

About Fig. R2.1: Yes, I see your point now. I was misinterpreting Figure 2, possibly because the warm colors of the MSLP gave me the impression of temperature. My apologies.

*Ok!*

This supplemental figure shows N-NE advection into the inland region. You might consider contouring the T-field so that advection is more easily seen. Also, you do not present any wind speed legend for the vector lengths so it is difficult for the reader to estimate an advection rate.

*Thank you for this suggestion, we tried the contouring (see Fig. 1), but believe that the additional visual noise, when combined with country borders, is a bit much, and that the country borders do help the orientation. We have included wind vector scales to the figures of the manuscript.*

[Figure]

*Fig 1: Example figure presenting with and without contoursfor the years 2004-2017.*

About the discussion of SST-FLC correlations on seasonal scales and your inclusion of Klein & Hartmann (1993) Fig. 6a in your response: You cite their showing SST dominating annual LTS changes, but you are arguing that LTS does \*not\* influence your day-to-day variability of FLC in your domain. Furthermore, the seasonal frequency of stratus from Klein & Hartmann (1993), with its distinct peak in SON, does \*not\* match your data set (their Fig. 4b). Perhaps that is because theirs is for 10-20°S, 0-10°E, a region downwind of yours.

*We agree that the comparison with Klein & Hartmann (1993) is not a perfect one, and now clarify this sentence: "In the adjacent marine regions downwind of the central Namibian coast, these stable conditions promote the formation of the southeastern Atlantic stratocumulus cloud deck and controls its seasonal cycle (Klein and Hartmann, 1993; Andersen et al., 2017), where the SST component is responsible for most of the LTS seasonality." The SST seasonality is similar in the two regions, however, as shown in Fig. 2 below.*

[Figure]

*Fig. 2: ERA5 SST monthly averages for the pixel 12°E, 23°S for the years 2004-2017.*

About the original comments surrounding p.9/l.10: I appreciate the extensive revisions you included in the discussion of TCWV, especially the references that attempt to quantify the longwave radiative impacts of the drier lower troposphere. However, I would suggest that your arguments with correlation should be taken with a grain of salt as they are by no means indicative of causation. This is particularly true in the case of water vapor and temperature, two meteorological variables that are highly correlated for strictly thermodynamic, and non-radiative, reasons. For example, I would not be surprised if a large portion of your Fig. R2.4 were not merely a manifestation of the ClausiusClapeyron relationship.

*We agree that in most regions, the Clausius-Clapeyron relationship is extremely relevant for the T2m-TCWV relationship. However, this is an arid environment and the TCWV anomalies that we observe are quite substantial, and can only be explained by advection, particularly as local evaporation will be low.*

About Figure R2.7: I appreciate the addition of the 3-panel (FLC, and clear, and difference) format of this new supplemental figure.

*Ok!*

About upwelling winds and SST response: Just to put a finer point on this SST lag to upwelling winds, from experience I expect that the lag time is something of order 5-15 hours in the midlatitudes. This is more like the period of an inertial oscillation, which is half of a pendulum day. In other words, I do not think you need to wait around for an equilibrium solution time scale of a couple of pendulum days to get the bulk of the temperature response to a wind shift. It is observed to occur much more rapidly. [See for example, Lentz, S.J., J. Phys. Ocean., 1992]

*Thanks for this comment. We acknowledge that the SST response is not binary and only switched on when reaching equilibrium, but rather a continuous change that features a time lag. We now state this more precisely on page 10, line 12 of the final version of the manuscript: "This is to be expected, as Ekman transport produces a steady-state situation only after a few pendulum days \ (Pond and Pickard, 2013), although an initial upwelling response can be expected earlier (Lentz, 1992)."*

About your subsidence reported in Figure 6: In regions of complex terrain and coastal geography I suspect that the low spatial resolution in the ERA5 reanalysis data probably does not capture subsidence very completely, but I realize it is the best you have available, and your other arguments are persuasive.

*Yes, we agree that some local scale effects are not resolved in ERA5.*

About your inclusion of potential temperature in Figure 8: I disagree that the primary difference is definitely the MBL Q (1.2-1.5 g/kg). The difference in potential temperature between clear and FLC days is ~2.5 K. At those temperatures the dQ/dT is about 0.7 g/kg/K, so those differences are comparable with respect to dew point depression.

*We agree, and have included this in the updated version of the manuscript on page 17, lines 5,6: "It is apparent that these air masses contain significantly more moisture and feature significantly lower Pot. T on FLC days than on clear days, which explains most of the difference in RH."*

In conclusion, I would like to thank the authors for their thorough and thoughtful responses to my review and for putting the significant effort into advancing the conversation and my understanding of fog and low cloud in these upwelling systems. Excellent job!

*Thank you, we appreciate the time and effort your spent on the reviews*

[revised manuscript text omitted]